inorganic chemistry/computational chemistry/synthetic chemistry

aminothiazole ligands, transition metal complexes, computational calculations, antibacterial, antioxidant activity

**Authors for correspondence:**
Sajjad Hussain Sumrra
e-mail: sajjadchemist@gmail.com
Muhammad Imran
e-mail: imranchemist@gmail.com

This article has been edited by the Royal Society of Chemistry, including the commissioning, peer review process and editorial aspects up to the point of acceptance.

# Metal incorporated aminothiazole-derived compounds: synthesis, density function theory analysis, *in vitro* antibacterial and antioxidant evaluation

Sajjad Hussain Sumrra[1], Zunaira Arshad[1],
Wardha Zafar[1], Khalid Mahmood[2], Muhammad Ashfaq[1],
Abrar Ul Hassan[1], Ehsan Ullah Mughal[1], Ahmad Irfan[3,4]
and Muhammad Imran[4]

[1]Department of Chemistry, University of Gujrat, Gujrat 50700, Pakistan
[2]Institute of Chemical Sciences, Bahauddin Zakariya University, Multan 60800, Pakistan
[3]Research Center for Advanced Materials Science, and [4]Department of Chemistry, Faculty of Science, King Khalid University, P.O. Box 9004, Abha 61413, Saudi Arabia

SHS, 0000-0002-1581-5451; EUM, 0000-0001-9463-9398

The present study advocates the combined experimental and computational study of metal-based aminothiazole-derived Schiff base ligands. The structure and electronic properties of ligands have been experimentally studied by spectroscopic methods (UV-Vis, FT-IR, [1]H-NMR and [13]C-NMR), mass spectrometry, elemental analysis and theoretically by density function theory (DFT). Computational calculations employing the B3LYP/6-31 + G(d,p) functional of DFT were executed to explore the optimized geometrical structures of ligands along with geometric parameters, molecular electrostatic potential (MEP) surfaces and frontier molecular orbital (FMO) energies. Global reactivity parameters estimated from FMO energy gaps signified the bioactive nature of ligands. The synthesized ligands were used for chelation with 3*d*-transition metals [VO(IV), Cr(III), Fe(II), Co(II), Ni(II), Cu(II) and Zn(II)] in 1 : 2 (metal : ligand) molar ratio. The spectral and magnetic results confirmed the formation of octahedral geometry around all the divalent and trivalent metal centres, whereas the tetravalent vanadyl centres were confirmed to have square-pyramidal geometry. All the as-synthesized compounds were investigated for *in vitro* antibacterial

potential against two Gram-negative (*Salmonella typhimurium* and *Escherichia coli*) and two Gram-positive (*Bacillus subtilis* and *Staphylococcus aureus*) bacteria. Antibacterial assay results displayed pronounced activity, and their activity is comparable to that of a standard drug (streptomycin). The antioxidant potential of these compounds was assessed by employing diphenyl picryl hydrazide radical scavenging activity. The results displayed that all the metal chelates have exhibited more bioactivity in contrast with free ligands. The chelation was the main reason for their enhanced bioactivity. These results indicated that the thiazole metal-based compounds could be exploited as antioxidant and antimicrobial candidates.

# 1. Introduction

Transition metal chelates of Schiff base ligands are of prodigious interest in the field of coordination chemistry [1]. Such metal complexes have attained special attention owing to their sensitivity, synthetic flexibility in addition to selectivity towards various metal ions [2]. Transition metal complexes were proved as more advantageous and applicable in comparison with conventional organic-based drugs because the pathogens have developed resistance against these drugs [3]. Likewise, these complexes have also been extensively investigated as bioactive compounds because of their wide-ranging biological activities [4]. In bio-inorganic chemistry, an exceptional concern in the Schiff base metal chelates is that they are capable of providing the synthetic models for the metal-holding sites in metalloenzymes/proteins owing to their fundamental role in the cancer diagnosis, tumour treatment, radio immunotherapy and clinical therapy [5]. Most of the organic compounds exhibit only one- or two-dimensional topologies, whereas metal complexes have the potential that they can create three-dimensional structures using the coordination chemistry of metal centres [6]. Thus, metal complexes are capable to develop a widespread range of antimicrobial agents. Moreover, metal complexes also have the ability to process antimicrobial actions in a variety of ways such as inhibition of microbial cell wall synthesis, disruption of microbial membrane, inhibition of nucleic acid synthesis, inhibition of protein synthesis [7], loss of vital substrates, release or exchange of ligands, interruption of membrane functions, denaturation of proteins, redox activation, destruction of DNA, abolishing the enzymatic activities as well as catalytic generation of some toxic species (reactive oxygen species, ROS) [8].

Metal ions play a significant role in all types of biological activities, as well as their rational design can also be used for developing new diagnostic probes or pharmaceutic drugs. The fact that metal atoms can readily give off electrons and produce cations assists their dissolution in biological fluids. Because of their electron-deficient nature, they can easily interact with proteins of DNA, which are electron-rich biomolecules, thus, participating either in the determination/stabilization of their tertiary or quaternary structures or in a catalytic mechanism [9]. There exist different mechanisms of antioxidant activity depending on the interaction of different ROS with metal complexes as antioxidant agents; interaction either with lipid hydroperoxides or lipid peroxyl radicals, thus terminating the basic chain radical process of lipid peroxidation, scavenging of hydroxyl radical as well as the dismutation of superoxide radical anion. The metal ions delocalize the electronic charge distribution of ligands by stabilizing their electronic system. This effect increases the antioxidant profile of ligands [10]. During recent years, Schiff bases synthesized from a variety of heterocyclic derivatives along with their metal coordinated complexes have attracted the interest of many researchers [11]. The strong aromatic nature of the heterocyclic ring system like different types of thiazoles and their derived compounds leads to various significant biological applications as well as causes an increase in their *in vivo* stability [12].

Thiazole consists of five-membered heterocyclic ring having nitrogen and sulfur at first and third positions in the ring system [13]. The biological significance of the thiazole ring is well recognized as well as reported in the literature, such as penicillin is the earliest known antibiotic drug having a tetrahydrothiazole moiety in its structure [14]. Likewise, vitamin B1 and the coenzyme cocarboxylase also contain a thiazole ring. Metal-based thiazole-derived compounds display a variety of biological activities including anti-inflammatory, antioxidant and antitubercular [15]. These compounds have also been emerged as a novel class of strong antimicrobial marketers, which could be reported for inhibiting the bacteria with the help of hindering off the biosynthetic pathway of positive bacterial lipids and/or by other additional mechanism [16]. Density functional specifications for designing the structure activity interaction are important for the study of molecular interaction layout, activity sites including biochemical activity [17]. In order to investigate the pharmacological and biological possibilities, it was important to focus on numerous molecular classifications such as molecular

electrostatic potential (MEP), frontier molecular orbitals (FMOs), electron affinity (EA) and ionization potential (IP) for the discovery of binding sites in bioactive compounds. The literature survey revealed that most of these compounds have shown a prominent improvement in the toxicological and pharmacological properties when coordinated with metal ions [18].

It has also been found that 2-aminothiazole compounds are generally more effective with a broad range of biological activity and are the basic intermediates formed during the synthesis of various pharmaceutical drugs [19]. Aminothiazole-derived compounds act as oestrogen receptors and represent a new class of adenosine receptor antagonists. Aminothiazole compounds also form Schiff base ligands after condensation with carbonyl compounds and exist in enolic as well as keto form [20], which are of special importance because of their extensive spectrum of biological applications including antimicrobial, antidiabetic [21], antiplatelet [22], antiviral, anti-cancer [23], antileishmanial, antianoxic, antituberculosis [24], anti-inflammatory [25], antihypertensive, antioxidant, DNA binding and cleavage [26]. Aminothiazole-derived Schiff base ligands have robust coordinating capability with metals, and these metal complexes have attracted great attention as they can introduce novel reactivity as well as stabilize the clusters of metal framework [27]. In the continuation of our ongoing research work on exploring the effective role of chelation on bioactive molecules [28], we herein report a new class of thiazole Schiff bases derived by the condensation reaction of two aminothiazole moieties with isatin and chloroisatin along with their transition metal complexes to scrutinize their antioxidant and antibacterial properties. The as-synthesized metal-based thiazole Schiff base ligands were characterized by computational and spectral studies, elemental analysis, magnetic and conductance measurements.

# 2. Experimental

## 2.1. Analytical and physical measurements

Analytical grade chemicals and reagents used in this research work were purchased from well-reputed chemical industries, Merck and Sigma-Aldrich, and used directly in the entire practice. The solvents that were used in the synthetic reactions were purified by the double distillation process. For the synthesis of compounds, a hotplate of Velp Scientifica was used for stirring and heating. Stuart melting point apparatus was used to record the melting points of the thiazole ligands as well as decomposition temperatures of the metal complexes. JEOL MS Linear Ion Trap Mass Spectrometer having an electrospray ionization probe was used to document the mass (LC–MS) spectra of the thiazole ligands. Nicolet FT-IR Impact 400D infrared spectrometer was used for recording IR spectra of the solid samples by the matrix of KBr in the range of 500–4000 $cm^{-1}$. A high-tech Shimadzu UV-4000 Spectrophotometer was employed for the UV-Vis analysis of all the synthesized compounds. Bruker Advance 300 MHz instrument was used to determine the $^{13}$C-NMR and $^{1}$H-NMR spectra using dimethyl sulfoxide (DMSO-$d_6$) as a solvent for making the solutions and internal standard of tetramethylsilane. By using Inolab Cond 720 Conductivity Bridge, molar conductivities of metal complexes were determined at room temperature by using $10^{-3}$ M sample solutions in DMF. Magnetic susceptibility balance was used to determine the magnetic susceptibilities of all the synthesized metal chelates at room temperature, using the standard of $MnCl_2$. All the biological activities were performed at the Department of Biochemistry, University of Gujrat, Gujrat, Pakistan.

## 2.2. Synthesis of ligands (Z₁)–(Z₃)

The thiazole-derived Schiff base ligands 3-[(6-ethoxy-1,3-benzothiazol-2-yl)imino]-1,3-dihydro-2*H*-indol-2-one (**Z₁**), 5-chloro-3-[(6-ethoxy-1,3-benzothiazol-2-yl)imino]-1,3-dihydro-2*H*-indol-2-one (**Z₂**) and 5-chloro-3-[(6-nitro-1,3-benzothiazol-2-yl)imino]-1,3-dihydro-2*H*-indol-2-one (**Z₃**) were prepared by using the equimolar ratio of 6-substitutedphenyl-aminothiazoles and aromatic ketones using an already reported protocol [29] as shown in scheme 1. The ligand (**Z₁**) was synthesized by adding an ethanolic solution of 2-amino-6-ethoxybenzothiazole (1.94 g, 10 mM) to a hot magnetically refluxed ethanolic solution of isatin (1.47 g, 10 mM) with constant heating and stirring. The reaction mixture was incessantly refluxed for 9 h. To check the progression of the reaction, comparative thin-layer chromatography (TLC) was taken after regular intervals of time. During the refluxing, orange coloured precipitates were formed that indicated the formation of ligand. Then, after cooling the reaction mixture to normal temperature, the precipitates were filtered, rinsed thoroughly with warm ethanol and then air-dried. The precipitates were recrystallized using the equimolar ratio of ethanol

**Scheme 1.** Synthetic scheme of aminothiazole Schiff bases **(Z$_1$)**–**(Z$_3$)** and their corresponding transition metal complexes **(1)**–**(21)**.

and ether. Then, the same protocol was applied for synthesizing the other two ligands **(Z$_2$)** and **(Z$_3$)** by refluxing the ethanolic solution of 2-amino-6-ethoxybenzothiazole and 2-amino-6-nitrobenzothiazole with 5-chloroisatin, respectively.

### 2.2.1. 3-[(6-Ethoxy-1,3-benzothiazol-2-yl)imino]-1,3-dihydro-2*H*-indol-2-one **(Z$_1$)**

Yield: 82% (2.65 g); M.P. (°C): 133; colour: orange; IR (KBr, cm$^{-1}$): 3149 (NH), 1725 (C=O), 1633 (C=N, azomethine), 1613 (C=N, thiazole), 1388 (C-O, ester), 862 (C-S); $^1$H-NMR (300 MHz, DMSO-$d_6$), δ (ppm): 11.05 (s, N-H), 7.58 (t, C$_7$-H), 7.49 (d, $J$ = 7.2 Hz, C$_{16}$-H), 7.26 (dd, $J$ = 7.2, 2.1 Hz, C$_{17}$-H), 7.22 (d, $J$ = 2.1 Hz, C$_{19}$-H), 7.06 (t, C$_8$-H), 6.92 (d, C$_9$-H), 6.77 (d, C$_6$-H), 3.97 (q, C$_{21}$-H), 1.33 (t, C$_{22}$-H); $^{13}$C-NMR (300 MHz, DMSO-$d_6$), δ (ppm): 184.86 (C$_2$), 165.15 (C$_{12}$), 159.83 (C$_4$), 153.92 (C$_{18}$), 151.17 (C$_{15}$), 147.22 (C$_{14}$), 138.82 (C$_{10}$), 132.33 (C$_8$), 125.15 (C$_7$), 123.21 (C$_{16}$), 118.51 (C$_5$), 118.28 (C$_9$), 113.84 (C$_{17}$), 112.65 (C$_6$), 106.64 (C$_{19}$), 63.96 (C$_{21}$), 15.24 (C$_{22}$); EIMS (70 eV) *m/z* (%): 324 (12), 310 (21), 291 (38), 270 (24), 247 (63), 232 (16), 203 (20), 192 (100), 171 (69) 152 (39), 134 (30), 120 (41), 95 (10), 81 (25), 76 (09), 43 (11); Anal. cald. for C$_{17}$H$_{13}$N$_3$O$_2$S (323.37) (%): C (63.14), H (4.05), N (12.99); found: C (63.51), H (4.16), N (13.15).

### 2.2.2. 5-Chloro-3-[(6-ethoxy-1,3-benzothiazol-2-yl)imino]-1,3-dihydro-2*H*-indol-2-one **(Z$_2$)**

Yield: 89% (3.19 g); M.P. (°C): 143; colour: rust; IR (KBr, cm$^{-1}$): 3148 (NH), 1749 (C=O), 1633 (C=N, azomethine), 1617 (C=N, thiazole), 1388 (C-O, ester), 862 (C-S), 819 (C-Cl); $^1$H-NMR (300 MHz, DMSO-$d_6$), δ (ppm): 11.15 (s, N-H), 7.62 (d, $J$ = 8.4 Hz, C$_{16}$-H), 7.59 (dd, $J$ = 8.4, 2.2 Hz, C$_{17}$-H), 7.55 (d, $J$ = 2.2 Hz, C$_{19}$-H), 7.23 (dd, $J$ = 8.1, 2.1 Hz, C$_8$-H), 6.91 (d, $J$ = 8.1 Hz, C$_9$-H), 6.77 (d, $J$ = 2.1 Hz, C$_6$-H), 3.96 (q, C$_{21}$-H), 1.33 (t, C$_{22}$-H); $^{13}$C-NMR (300 MHz, DMSO-$d_6$), δ (ppm): 183.81 (C$_2$), 165.17 (C$_{12}$), 159.64 (C$_4$), 153.93 (C$_{18}$), 149.65 (C$_{15}$), 147.14 (C$_{14}$), 137.69 (C$_{10}$), 132.29 (C$_8$), 127.24 (C$_7$), 124.60 (C$_{16}$), 119.62 (C$_9$), 118.49 (C$_5$), 114.28 (C$_6$), 113.84 (C$_{17}$), 106.64 (C$_{19}$), 63.95 (C$_{21}$), 15.24 (C$_{22}$); EIMS (70 eV) *m/z* (%): 358 (06), 329 (12), 313 (23), 293 (39), 282 (10), 266 (23), 196 (10), 194 (80), 168 (08), 165 (100), 140 (07), 139 (29), 110 (22), 95 (08), 80 (11), 69 (17) 45 (12); Anal. cald. for C$_{17}$H$_{12}$ClN$_3$O$_2$S (357.81) (%): C (57.06), H (3.38), N (11.74); found: C (57.29), H (3.67), N (11.24).

### 2.2.3. 5-Chloro-3-[(6-nitro-1,3-benzothiazol-2-yl)imino]-1,3-dihydro-2*H*-indol-2-one **(Z$_3$)**

Yield: 89% (3.21 g); M.P. (°C): 219; colour: mustard; IR (KBr, cm$^{-1}$): 3075 (NH), 1747 (C=O), 1653 (C=N, azomethine), 1612 (C=N, thiazole), 1322 (NO$_2$), 845 (C-S), 826 (C-Cl); $^1$H-NMR (300 MHz, DMSO-$d_6$), δ (ppm): 11.14 (s, N-H), 8.68 (d, $J$ = 2.0 Hz, C$_{19}$-H), 8.25 (dd, $J$ = 8.1, 2.0 Hz, C$_{17}$-H), 7.59 (d, $J$ = 8.1 Hz, C$_{16}$-H), 7.54 (d, $J$ = 2.2 Hz, C$_6$-H), 7.42 (dd, $J$ = 8.4, 2.2 Hz, C$_8$-H), 6.93 (d, $J$ = 8.4 Hz, C$_9$-H); $^{13}$C-NMR

(300 MHz, DMSO-$d_6$), $\delta$ (ppm): 183.80 ($C_2$), 172.25 ($C_{12}$), 159.63 ($C_4$), 159.06 ($C_{18}$), 149.65 ($C_{15}$), 141.10 ($C_{14}$), 137.69 ($C_{10}$), 132.03 ($C_8$), 127.24 ($C_7$), 124.59 ($C_{16}$), 122.46 ($C_{17}$), 119.61 ($C_9$), 118.18 ($C_5$), 117.30 ($C_{19}$), 114.27 ($C_6$); EIMS (70 eV) $m/z$ (%): 359 (06), 331 (36), 197 (81), 165 (78), 149 (24), 137 (20), 122 (100), 105 (17), 95 (22), 78 (16), 63 (80), 51 (08), 45 (21); Anal. cald. for $C_{15}H_7ClN_4O_3S$ (358.76) (%): C (50.22), H (1.97), N (15.62); found C (50.66), H (1.60) N (15.98).

## 2.3. Synthesis of transition metal complexes (1)–(21)

All the transition metal complexes were synthesized using ligands and metallic salts in 2 : 1 molar ratio. For this purpose, the ethanolic solution of respective transition metallic salt (5 mmol) was lowered dropwise to the hot magnetically refluxed ethanolic solution of thiazole Schiff base ligand (10 mmol) in two-neck round-bottom flask with continual stirring and heating. Then, the resultant mixture was magnetically stirred under reflux for 6 to 7 h, during which a precipitated metal complex was formed. The product formation was designated by the TLC of the reaction mixture. The coloured precipitates of the product were filtered, rinsed with hot ethanol and then dried. Likewise, all the other metal complexes (1)–(21) have also been prepared using the same procedure.

## 2.4. Computational details

All the quantum chemical calculations for the investigated thiazole ligands as well as their selected 3$d$-metal complexes were accomplished by density function theory (DFT) exploiting Gaussian 09 software package [30]. The geometrical structures of the ligands were optimized without any symmetry constraints by applying B3LYP level of DFT with the combination of standard double zeta plus polarization basis set 6-31 + G(d,p) and LanL2DZ. The B3LYP method of DFT study is quite applicable for predicting the electronic as well as geometric properties of charged and neutral chemical systems extending from simple molecular structures to polymeric structures. The input files were organized using GaussView software [31]. The Chemcraft [32], GaussView, GaussSum [33] and Avogadro [34] programs were used for interpreting the output files. The key purpose of these quantum mechanics calculations is to validate the predicted three-dimensional structure of the studied ligands in addition to find the main aspects for their activities.

## 2.5. Antibacterial activity

The as-synthesized metal-based thiazole Schiff base ligands were scrutinized for developing new antibacterial agents. The growth-inhibiting potentials of all the compounds were explored against two Gram (+) bacteria; *Bacillus subtilis* and *Staphylococcus aureus* as well as two Gram (−) bacteria; *Salmonella typhimurium* and *Escherichia coli* by standard disc diffusion method [35]. Streptomycin was employed as the standard antibacterial agent in this assay. The materials used in the antibacterial activity assay such as nutrient broth; Petri plates as well as discs of filter paper were autoclaved for 30 min at 121°C. The solutions of standard drug and tested compounds were prepared in DMSO solvent to obtain the desired concentrations (5 mg ml$^{-1}$). DMSO (100 µl) was used as the negative control, while the standard drug was used as a positive control. The bacterial culture was mixed in the nutrient broth, then the resultant mixture was evenly spread on the Petri plates. After solidification, the sterilized filter paper discs were saturated with 100 µl of the standard and samples. After that, the Petri plates were kept for incubation at 37°C temperature for 48 h. At the end, clear or inhibition zones were noted (in mm) for all the tested compounds and standard drug against each bacterial strain.

## 2.6. Antioxidant activity

The antioxidant action of all the synthesized compounds was accomplished through diphenyl picryl hydrazide (DPPH) radical scavenging method. The detail of the method is described below.

### 2.6.1. Diphenyl picryl hydrazide radical scavenging method

The antioxidant potential of all the synthesized compounds and the standard antioxidant agent, butylated hydroxytoluene (BHT), was evaluated on the base of the radical scavenging aptitude of 1,1-diphenyl-2-picryl hydrazyl (DPPH) stable free radical. The radical scavenging actions of antioxidant compounds on the DPPH radical are owing to their ability to donate hydrogen atom which results in

the decrease in absorbance reading at 517 nm wavelength. The standard procedure as described in the literature was adopted in this assay [36]. For this purpose, 20 mg/100 ml solution of DPPH radical and sample solutions of the synthesized compounds in two different concentration levels (1 and 2 mg ml$^{-1}$) were prepared. The reaction mixture was prepared in test tubes by adding 2 and 4 ml of methanolic solution of DPPH in 1 and 2 ml of sample solutions, respectively. Then, the test tubes were covered in aluminium foil and kept in an incubator at 37°C for half an hour. At 517 nm wavelength, the absorption of the prepared compounds was determined and the percentage inhibition values were calculated by using the formula given in equation (2.1),

$$\text{Percentage (\%) inhibition} = \frac{(\text{Blank} - \text{Sample})}{\text{Blank}} \times 100. \tag{2.1}$$

# 3. Results and discussion

In this study, series of thiazole Schiff base ligands $(Z_1)$–$(Z_3)$ were prepared by refluxing ethanolic solutions of 2-amino-6-ethoxybenzothiazole and 2-amino-6-nitrobenzothiazole with isatin and chloroisatin in equimolar concentration by employing a condensation reaction mechanism (scheme 1). The structures of the ligands were elucidated using physical, spectroscopic, elemental and computational analysis. The synthesized ligands were soluble in majority of the organic solvents like ethanol, methanol, DMSO, dichloromethane, dioxane and acetone. In addition, these synthesized ligands were used to derive the complexes of metals [VO(IV), Cr(III), Fe(II), Co(II), Ni(II), Cu(II) and Zn(II)] in a stoichiometric ratio (2 L : 1 M). Vanadium, iron and zinc metals were used as their sulfates; chromium was used as acetate while cobalt, nickel and copper were used as their chloride salts.

All the powdered metal-based compounds were non-hygroscopic, stable at room temperature and intensely coloured except zinc chelates. All the as-synthesized compounds were prepared in good to excellent yield ranging from 75 to 91%. Melting/decomposition points and the thiazole-derived ligands and complexes gave a strong clue regarding the formation of both ligands and their corresponding metal complexes. The spectral data indicated that the ligands were coordinated with the 3*d*-metal ions by means of bidentate and tridentate approach using NN and NNO donor atoms. The physical properties in addition to the microanalytical details of the synthesized thiazole Schiff base ligands and their respective metal-based compounds are depicted in the electronic supplementary material, table S1.

## 3.1. FT-IR spectra

All the Schiff base ligands have shown the disappearance of vibrational band at 3310 cm$^{-1}$, developing into the band of a new functional group, azomethine $v$(C=N) at 1633–1653 cm$^{-1}$, giving the confirmation of the condensation of the amino group of the aminothiazole moiety with one of the carbonyl group of isatin [37]. The significant vibrational bands of the thiazole Schiff base ligands and their corresponding transition metal complexes are detailed in the electronic supplementary material, table S2, while their FT-IR spectra are shown in the electronic supplementary material, figures S1–S9. All the ligands have shown strong vibrational bands at 845–862, 1612–1617, 1725–1749 and 3075–3149 cm$^{-1}$ due to $v$(C-S), $v$(C=N) of the thiazole ring [38], aromatic carbonyl $v$(C=O) and secondary amine group $v$(N-H) of the isatin moiety, respectively. The $v$(C=C) and $v$(C-H) bands appeared at 1540–1568 and 2928–2974 cm$^{-1}$ in all the ligands spectra. Moreover, the ligands $(Z_1)$ and $(Z_2)$ showed the vibrational band at 1388 cm$^{-1}$ owing to ester $v$(C-O) group [39]. And ligands $(Z_2)$ and $(Z_3)$ displayed a band at 819–826 cm$^{-1}$, as a result of $v$(C-Cl). The band at 1322 cm$^{-1}$ was specified for the $v$(NO$_2$) group of the ligand $(Z_3)$ [40].

The IR spectral data confirmed the coordination of metals with respective ligands through coordinated sites: oxygen of isatin, nitrogen of azomethine and nitrogen of thiazole moiety. The vibrational band present at 1633–1653 cm$^{-1}$ in the IR spectra of the ligands owing to azomethine linkage was moved to a lower frequency at 1615–1635 cm$^{-1}$ displaying the contribution of the nitrogen of azomethine group in the complex formation. The $v$(C=N) vibrational band of aminothiazole linkage was also shifted from 1612–1617 cm$^{-1}$ to lower frequency at 1602–1607 cm$^{-1}$, representing the presence of M-N bond in all the metal complexes [41]. The $v$(C=O) vibrational bands of isatin moiety have also shifted from 1725–1749 to 1703–1728 cm$^{-1}$, signifying the coordinating action of the oxygen of isatin moiety with the metal ions. The occurrence of band above 527 cm$^{-1}$ represented the coordinating action of oxygen of isatin moiety with metal ions because of $v$(M–O)

group vibrations [42]. The vibrational bands at 2928–2974, 845–862, 1388 and 1540–1568 cm$^{-1}$ allocated to aromatic $v$(C–H), $v$(C–S), $v$(C–O) and $v$(C=C), correspondingly in spectra remained unchanged predicting their no participation in the complexation. All the oxovanadium(IV) complexes exhibited vibrational band at 940–972 cm$^{-1}$ that signified the incorporation of the $v$(V=O) group of vanadyl(IV). These complexes also possessed a vibrational band at 1053–1059 cm$^{-1}$ because of $v$(SO$_4$) group. All these indications confirmed the formation of thiazole Schiff base ligands and their resulting 3$d$-metal-based compounds [43].

## 3.2. $^1$H-NMR spectra

The $^1$H-NMR spectral data provided further support for the molecular structure and composition of the synthesized compounds. All the protons of the ligands and their diamagnetic Zn(II) complexes have displayed their signals in the expected regions because of the aromatic as well as heterocyclic groups. The values are given in the electronic supplementary material, tables S3 and S4 for ligands **(Z$_1$)**–**(Z$_3$)** and their derived zinc complexes **(1)** and **(14)**, respectively [44]. The singlet and doublet signals appeared in the spectra of all the ligands **(Z$_1$)**–**(Z$_3$)** in the range 11.05–11.15 and 6.91–6.93 ppm due to N-$H$ and C$_9$-$H$, respectively (electronic supplementary material, figures S10–S12). For ligands **(Z$_1$)** and **(Z$_2$)**, signals for the methylene (C$_{21}$-$H$) and methyl (C$_{22}$-$H$) protons appeared at 3.96–3.97 and 1.33 ppm as quartet and triplet, respectively. The C$_8$-$H$ and C$_6$-$H$ protons appeared at 7.06 ppm and 6.77 ppm as triplet and doublet, respectively, for ligand **(Z$_1$)**, while for the other two ligands **(Z$_2$)** and **(Z$_3$)**, these protons were observed downfield as a doublet of doublet and doublet in the range of 7.23–7.42 and 6.77–7.54 ppm, correspondingly because of the presence of electron-withdrawing chloro (Cl) group at the adjacent carbon (C$_7$) atom.

Similarly for C$_7$-$H$, a triplet signal was observed at 7.58 ppm in the ligand **(Z$_1$)**, whereas no signal was observed for other two ligands **(Z$_2$)** and **(Z$_3$)** as a result of the attachment of Cl group at that position. For all the ligands, the C$_{16}$-$H$ proton was observed in the range 7.49–7.62 ppm as a doublet signal. For ligands **(Z$_1$)** and **(Z$_2$)**, the aromatic protons C$_{17}$-$H$ and C$_{19}$-$H$ of benzothiazole moiety were observed in the range 7.26–7.59 and 7.22–7.55 ppm as doublet of doublet and doublet, respectively. In the case of ligand **(Z$_3$)**, because of the presence of more electron-withdrawing nitro (NO$_2$) group, the values of the protons C$_{17}$-$H$ and C$_{19}$-$H$ were deshielded and observed at 8.25 and 8.68 ppm as a doublet of doublet and singlet, correspondingly. The comparison of $^1$H-NMR spectra of diamagnetic Zn(II) complexes (electronic supplementary material, figure S13–S14) with the free ligands revealed that the proton signals underwent slight shift upon coordination due to increased conjugation.

## 3.3. $^{13}$C-NMR spectra

The $^{13}$C-NMR spectral analysis in DMSO solvent provided strong evidence about the formation of ligands **(Z$_1$)**–**(Z$_3$)** and their diamagnetic Zn(II) complexes. The $^{13}$C-NMR spectra displayed characteristic signals within the expected range. The values are listed in the electronic supplementary material, tables S5 and S6 for ligands **(Z$_1$)**–**(Z$_3$)** and their derived zinc complexes **(1)** and **(14)**, correspondingly. For all ligands **(Z$_1$)**–**(Z$_3$)**, the carbon C$_4$ signal of azomethine (C=N) appeared at 159.63–159.83 ppm that strongly supported the synthesis of ligands by a condensation reaction (electronic supplementary material, figures S15–S17). Because of the inductive effect of nitrogen and electronegative effect of oxygen, C$_2$ carbon was observed downfield at 183.80–184.86 ppm demonstrating the existence of C=O carbon. Carbon atom C$_{12}$ present between nitrogen and sulfur atom of aminothiazole exhibited signal at 165.15–172.25 ppm.

For ligand **(Z$_1$)**, the C$_7$ signal of the phenyl group of isatin appeared at 125.15 ppm, but for other two ligands **(Z$_2$)** and **(Z$_3$)**, it was shifted downfield and observed at 127.24 ppm due to the attachment of more electronegative chloro group. For ligands **(Z$_1$)** and **(Z$_2$)**, the carbon C$_{18}$ of the phenyl group of aminothiazole showed a major shift and appeared at 153.92–153.93 ppm due to the presence of oxygen atom (more electronegative), whereas for ligand **(Z$_3$)**, the signal for the same carbon atom was observed upfield at 122.46 ppm owing to the presence of a less electronegative nitro group. All the other carbon signals agreed well with their expected values as well as compromised well with the total number of carbon atoms existing in the suggested structures of the ligands. Slight shifting of the carbonyl carbon (C$_2$) and azomethine carbon (C$_4$) in the divalent zinc complexes revealed coordination of the carbonyl-O and azomethine-N to the zinc atom (electronic supplementary material, figures S18–S19).

## 3.4. Mass spectra

Mass spectra of the ligands **(Z₁)–(Z₃)** helped in determining their structure by means of charge to mass ratio (electronic supplementary material, figures S20–S22). In the mass spectrum of ligand **(Z₁)**, the molecular ion fragment with the molecular formula $[C_{17}H_{13}N_3O_2S]$ appeared at $m/z = 324$ and the base fragment of thiazole ligand was obtained at $m/z = 192$ agreeing well with the fragment $[C_9H_8N_2OS]$. The other significant molecular fragments along with their relative abundance were observed at 310 (21%), 291 (38%), 270 (24%), 247 (63%), 232 (16%), 203 (20%), 171 (69%) 152 (39%), 134 (30%), 120 (41%), 95 (10%), 81 (25%), 43 (11%) due to fragments $[C_{16}H_{12}N_3O_2S^{3\bullet}]$, $[C_{17}H_{13}N_3O_2^{\bullet}]$, $[C_{14}H_{10}N_2O_2S^{4\bullet}]$, $[C_{11}H_9N_3O_2S^{2\bullet}]$, $[C_{11}H_8N_2O_2S^{2\bullet}]$, $[C_9H_5N_3OS^{2\bullet}]$, $[C_9H_5N_3O^{2\bullet}]$, $[C_8H_8OS^{2\bullet}]$, $[C_8H_8NO^{\bullet}]$, $[C_8H_8O^{2\bullet}]$, $[C_3HN_3O^{4\bullet}]$, $[C_4H_3NO^{4\bullet}]$, $[CHNO^{2\bullet}]$, respectively. Some more molecular fragments for the ligand were observed at 284, 212, 184, 144, 109, 76 and 53 as a result of the fragmentation of C=N, C–S, C–N, C–O and C–C.

The molecular ion fragment of the ligand **(Z₂)**, $[C_{17}H_{12}ClN_3O_2S]$ was found to be at $m/z = 358$ and the most stable fragment $[C_8H_4ClNO]$ of maximum intensity (100%) at $m/z = 165$. Beside these, the most prominent molecular fragments were observed at 329 (12%), 313 (23%), 293 (39%), 282 (10%), 266 (23%), 196 (10%), 194 (80%), 139 (29%), 110 (22%), 80 (11%), 69 (17%) and 45 (12%) because of fragments $[C_{16}H_{12}ClN_3OS^{2\bullet}]$, $[C_{15}H_8ClN_3OS^{\bullet}]$, $[C_{13}H_{10}ClN_2O_2S^{3\bullet}]$, $[C_{14}H_8N_3O_2S^{3\bullet}]$, $[C_{14}H_8N_3OS^{3\bullet}]$, $[C_7H_4ClN_3S^{4\bullet}]$, $[C_8H_4ClN_2S^{4\bullet}]$, $[C_7H_7OS^{4\bullet}]$, $[C_6H_4Cl^{2\bullet}]$, $[C_5H_4O^{2\bullet}]$, $[C_3H_3NO^{3\bullet}]$ and $[C_2H_5O^{\bullet}]$, correspondingly. While the other molecular fragments of less than 10% relative abundance were observed at 250, 234, 213, 168, 140 and 95. The molecular fragment of the ligand **(Z₃)**, $[C_{15}H_7ClN_4O_3S]$ appeared at $m/z = 359$, while the most stable (base fragment) with the $[C_6H_4NO_2^{2\bullet}]$ formula was observed at $m/z = 122$. Some major molecular fragments having evident percentages were found at 331 (36%), 197 (81%), 165 (78%), 149 (24%), 137 (20%), 105 (17%), 95 (22%), 78 (16%), 63 (80%) and 45 (21%) owing to fragments $[C_{13}H_5ClN_4O_3S^{2\bullet}]$, $[C_7H_4ClN_3S^{4\bullet}]$, $[C_6H_2N_2O_2S^{\bullet}]$, $[C_5N_3OS^{6\bullet}]$, $[C_7H_4ClN^{3\bullet}]$, $[C_6HS^{3\bullet}]$, $[C_3HN_3O^{3\bullet}]$, $[C_5H_4N^{2\bullet}]$, $[C_4HN^{5\bullet}]$ and $[CH_3NO^{2\bullet}]$, respectively. While the other molecular fragments of less than 10% relative abundance were observed at 322, 304, 288, 266, 237, 211, 179 and 51. The aspects of the thiazole Schiff base ligands obtained by their mass spectra greatly validated their synthesis and agreed well with their suggested structures.

## 3.5. Electronic spectra

The molecular electronic spectral measurements are very significant for assigning the stereochemistry and geometrical arrangement to transition metal complexes on the basis of the sites as well as a number of d-d transitions. The electronic spectra of the thiazole ligands **(Z₁)–(Z₃)** and their corresponding transition metal complexes were obtained at 298 K in the wavelength range 200–800 nm using DMF solvent. The electronic spectra of the prepared ligands showed prominent bands at 253–298 nm due to $\pi \rightarrow \pi^*$ electronic transitions seemed within the aromatic ring. The other absorbance band recorded at 306–392 nm was attributed to $n \rightarrow \pi^*$ electronic transitions of azomethine linkage in the free ligands [45].

In the spectra of metal complexes, these two bands owing to $\pi \rightarrow \pi^*$ and $n \rightarrow \pi^*$ electronic transitions were moved to higher frequencies because of the chelation with metals. Apart from these two bands, metal complexes also exhibited bands due to d-d electronic transitions. The high-intensity absorption bands of the different metal complexes at $\lambda_{max} = 310$–380 nm were assigned to charge transfer from aminothiazole ligands to transition metal ions [46]. The bands observed for the metal complexes are depicted in the electronic supplementary material, table S2. Based on the data of electronic absorption studies, an octahedral geometry was allocated to all the metal-based compounds except oxovanadium(IV) complexes **(1), (8)** and **(15)** which have shown square-pyramidal geometry.

## 3.6. Molar conductance

The molar conductivity analysis of the metal complexes was performed for 0.0001 M solutions in DMF solvent at room temperature. High molar conductance values were observed for all the metal complexes ranging from 82 to 221 $\Omega^{-1} \, cm^2 \, mol^{-1}$, confirming their electrolytic nature [47]. All the complexes exhibited different conductance values depending on the number of anions present in their outer sphere (given in the electronic supplementary material, table S2). Cr(III) complexes have shown the highest conductance values due to the presence of three acetate ($CH_3COO^-$) ions, Co(II), Ni(II) and Cu(II) complexes exhibited moderate conductance values owing to the presence of two chloride ($Cl^-$) ions, whereas the least conductance values of VO(IV), Fe(II) and Zn(II) complexes were attributed to

sulfate ($SO_4^{-2}$) ion [48]. Overall, the decreasing order for molar conductance values was Cr(III) > Co(II), Ni(II), Cu(II) > VO(IV), Fe(II), Zn(II).

## 3.7. Magnetic susceptibility

The magnetic susceptibility evidenced very valuable in deciding the stereochemistry of 3*d*-metal cations in their chelates. Magnetic moment values help in predicting the paramagnetic and diamagnetic nature of metal chelates by providing the necessary information regarding the number of unpaired electrons in the d-orbitals of metal ions. This study also provides assistance in determining the geometry of metal chelates. The magnetic moment values of the synthesized transition metal chelates are depicted in the electronic supplementary material, table S2. The square-pyramidal geometry for the oxovanadium complexes **(1)**, **(8)** and **(15)** was verified by their experimental magnetic moments ranging from 1.70 to 1.76 Bohr's magneton (BM) signifying only one unpaired electron [49]. The chromium complexes **(2)**, **(9)** and **(16)** displayed magnetic moment values ranging from 3.82 to 3.91, confirming their octahedral geometry and paramagnetic nature owing to the d-orbital containing three unpaired electrons. The iron complexes **(3)**, **(10)** and **(17)** were found to be highly paramagnetic, exhibiting 4.84–4.92 BM magnetic moment values authorizing their octahedral geometry with four unpaired electrons. The cobalt complexes **(4)**, **(11)** and **(18)** have exhibited 4.29–4.39 BM magnetic moment values, describing these complexes to be paramagnetic owing to the presence of three unpaired electrons inferring their octahedral geometry. The magnetic moment values of 2.83–2.91 and 1.74–1.80 BM confirmed the octahedral symmetry of nickel complexes **(5)**, **(12)** and **(19)** and copper complexes **(6)**, **(13)** and **(20)** with two and one unpaired electrons, respectively, showing paramagnetic properties. The zero magnetic moment value for all the zinc complexes **(7)**, **(14)** and **(21)** confirmed their diamagnetic nature and octahedral geometry having no unpaired electron [50].

## 3.8. Geometry optimization studies

The optimized geometries of the synthesized thiazole ligands and their selected metal-based compounds were computed at their ground state energies ($S_0$) by using the DFT approach with the Gaussian 09 software. The DFT with B3LYP/6-31 + G(d,p) and B3LYP/6-31 + G(d,p)(LanL2DZ) basis sets have been used to refine geometry, respectively, so that ligands and their derived complexes can have the lowest energy configurations. The DFT coordinates of optimized compounds are given in the electronic supplementary material, tables S7–S12; whereas, the selected bond lengths and bond angles of the optimized compounds are depicted in the electronic supplementary material, table S13. The optimized geometrical structures of all the studied compounds were envisioned with Chemcraft software. Figure 1 represents the molecular structures of studied thiazole-derived compounds with atom numbering.

In DFT-based computational calculations, initially, the geometries of both thiazole ligands and their derived complexes were optimized without using any symmetry. Afterwards, the stability of optimized geometrical structures was confirmed by using the frequency calculations based on the same basis set. Out of all the theoretically obtained frequencies, no negative eigenvalue was found, which revealed that the achieved optimized geometrical structures of studied compounds were found at true minimal values in their potential energy surfaces. The structures of all studied compounds attained after geometry optimization were found to be neutral with a singlet spin state.

DFT optimized bonding distances, 1.42–1.41, 1.32–1.34, 1.33–1.38, 1.36–1.38, 1.32–1.38 and 1.33–1.38 Å, have been found via the benzene (C-C) system. Ligands have been shown to have enlarged lengths adapted for bonding of $C_7$-$N_{14}$ (1.27A) and $N_1$-$C_7$ (1.29 A) in the context of their $C_9$-$N_{10}$, $C_{11}$-$N_{13}$, $C_9$-$N_{10}$ and $C_{10}$-$N_{11}$ bonds, following the coordination of metal (V, Cr, Fe, Co Ni, Cu and Zn) of ligands. This situation declined the prestige of azomethine as a dual bond (C=N), which fostered chelation and has also influenced M-N bond development. The UV-visible analysis was accomplished using TD-DFT with the same basis set. However, the other computed analyses including IR, FMO, natural bond orbital (NBO), Mulliken atomic charge (MAC) and MEP were executed on these optimized geometrical structures using the same functional and basis set.

## 3.9. Frontier molecular orbital analysis

The electronic properties of a chemical system as well as its ability to absorb light could be simply estimated by employing the FMO analysis. The kinetic stability and chemical reactivity of any

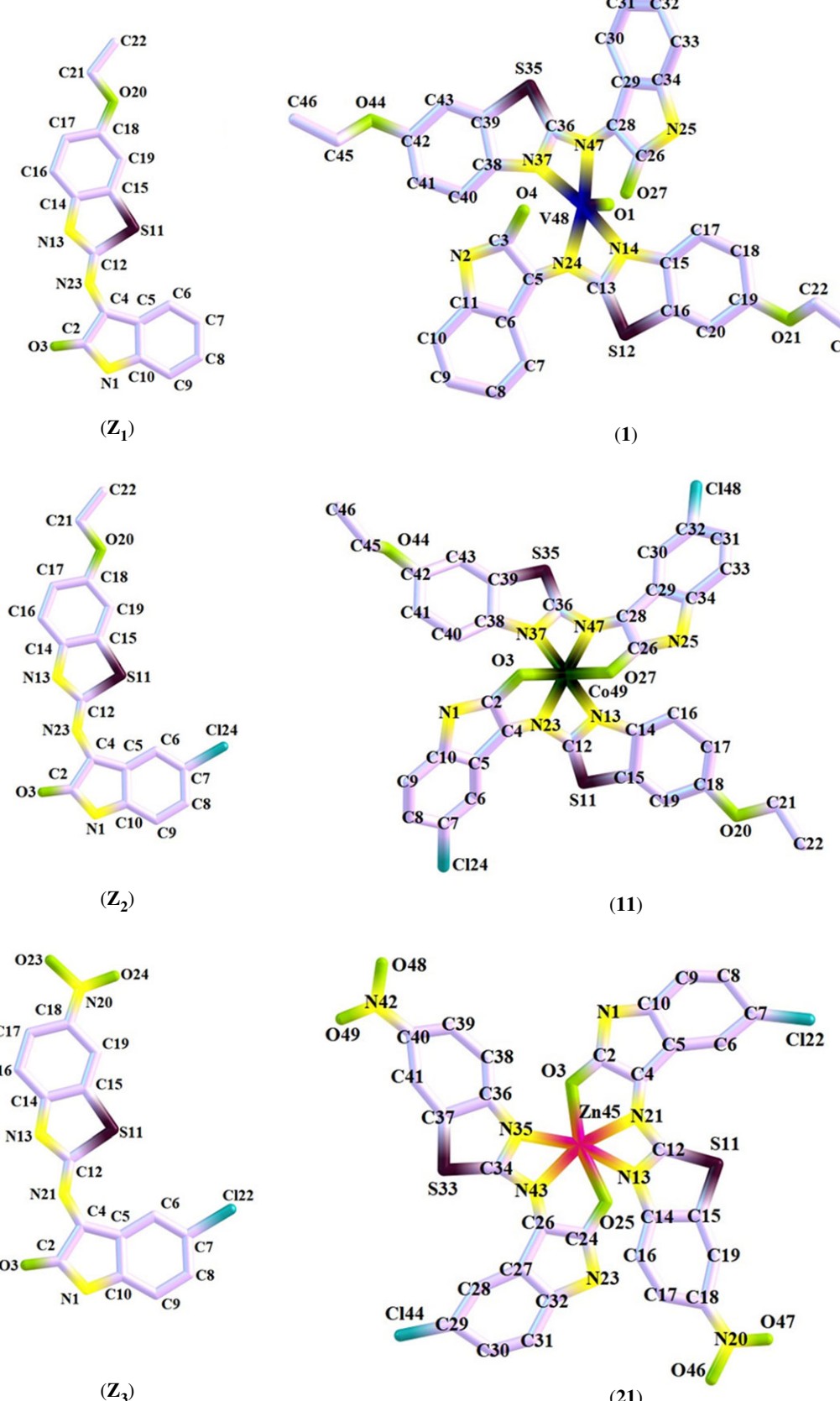

**Figure 1.** Optimized geometrical structures of ligands (**Z₁**)–(**Z₃**) and their selected metal-based compounds (**1**), (**11**) and (**21**).

chemical compound is directly associated with the difference of $E_{HOMO}$ and $E_{LUMO}$ [51]. The ability to accept electron is the feature of LUMO orbital which is electronically empty or unoccupied orbital while the ability to donate electron is the feature of HOMO orbital which is filled with electrons. The

**Table 1.** Calculated FMO energies ($E$), energy gap ($\Delta E$), ionization potential ($IP$), electron affinity ($EA$), chemical potential ($\mu$), global hardness ($\eta$), global softness ($\sigma$), global electrophilicity ($\omega$) and electronegativity ($\chi$) of reference compounds, thiazole ligands ($Z_1$)–($Z_3$) and their selected metal complexes **(1)**, **(11)** and **(21)**.

| descriptor (eV) | gallic acid | ascorbic acid | indomethacin | ($Z_1$) | ($Z_2$) | ($Z_3$) | (1) | (11) | (21) |
|---|---|---|---|---|---|---|---|---|---|
| $E_{HOMO}$ | −6.44 | −6.71 | −6.00 | −5.53 | −5.66 | −6.57 | −7.35 | −6.75 | −7.35 |
| $E_{HOMO-1}$ | −6.76 | −8.03 | −6.17 | −6.21 | −6.28 | −6.96 | −7.37 | −6.78 | −7.43 |
| $E_{LUMO}$ | −1.63 | −1.16 | −2.16 | −2.75 | −2.94 | −3.58 | −7.07 | −6.56 | −7.07 |
| $E_{LUMO+1}$ | −0.56 | −0.45 | −1.26 | −0.30 | −0.58 | −2.27 | −6.99 | −5.74 | −7.16 |
| $\Delta E_{H-L}$ | 4.84 | 5.55 | 3.84 | 2.78 | 2.72 | 2.99 | 0.27 | 0.19 | 0.27 |
| $\Delta E_{H-1-L+1}$ | 6.20 | 7.58 | 4.91 | 5.91 | 5.70 | 4.69 | 0.38 | 1.03 | 0.29 |
| $IP$ | 6.44 | 6.71 | 6.00 | 5.53 | 5.66 | 6.57 | 7.35 | 6.75 | 7.35 |
| $EA$ | 1.63 | 1.16 | 2.16 | 2.75 | 2.94 | 3.58 | 7.07 | 6.56 | 7.07 |
| $\eta$ | 2.40 | 2.54 | 1.92 | 1.39 | 1.36 | 1.49 | 7.21 | 6.65 | 7.21 |
| $\mu$ | −4.03 | −4.17 | −4.08 | −4.14 | −4.30 | −5.07 | −7.21 | −6.65 | −7.21 |
| $\sigma$ | 1.34 | 1.31 | 1.56 | 1.99 | 2.08 | 2.19 | 0.14 | 0.10 | 0.14 |
| $\chi$ | 4.03 | 4.17 | 4.08 | 4.14 | 4.30 | 5.07 | 3.67 | 5.25 | 3.67 |
| $\omega$ | 3.39 | 3.42 | 4.33 | 6.16 | 6.80 | 8.64 | 19.09 | 22.39 | 11.09 |

small difference of $E_{HOMO}$ and $E_{LUMO}$ is related to more polarizing ability of the molecule and more chances of intramolecular charge transfer (ICT) and vice versa [52].

The energies of FMOs including HOMO−1, HOMO, LUMO, LUMO+1 along with HOMO–LUMO energy differences ($\Delta E_{HOMO-LUMO}$) and HOMO−1–LUMO+1 energy differences ($\Delta E_{HOMO-1-LUMO+1}$) are significant characteristic features for exploring the electronic properties of chemical systems. In this study, B3LYP/6-31 + G(d,p) basis set was employed to calculate the energies of FMOs, i.e. HOMO, HOMO−1, LUMO, LUMO+1 and the differences of their energies $\Delta E_{HOMO-LUMO}$ and $\Delta E_{HOMO-1-LUMO+1}$ for reference compounds, all the three ligands and their selected metal complexes as tabulated in table 1 and illustrated in figure 2. Figure 3 displays the HOMO–LUMO of complexes **(1)**, **(11)** and **(21)**, while HOMO–LUMO of all the other compounds are shown in the electronic supplementary material, figures S23–S25.

Previous studies revealed that gallic acid, indomethacin and ascorbic acid are potent drugs and used as reference candidates. It was also found that gallic acid [53], indomethacin [54] and ascorbic acid [55] have been proved effective drugs like antimicrobial, antioxidant and antibiotics. Thus in the current study, we have compared the nature of biological activity of our synthesized compounds with these reference drugs. The antioxidant aptitude and biological action of thiazole ligands are also strongly linked to the spatial spreading of occupied molecular orbitals specifying the most probable positions in the studied ligands which can be surely attacked by reactive agents in addition to free radicals. The FMOs are overlapped, signifying the tremendously reactive behaviour of the ligands towards antibacterial and antioxidant activity [56]. The compounds having reduced $E_{HOMO}$ values normally exhibited weaker electron-donating capability exemplifying that ligand **($Z_1$)** and **($Z_2$)** may have better electron-donating aptitude in contrast with reference compounds resulting in these ligands would be enhanced antioxidant and biological active contenders.

The work functions of Al and Au are 4.08 and 5.10 eV, respectively [57]. The hole/ electron injection energies (HIE/EIE) of thiazole ligands and standard compounds to Al and Au electrodes are investigated. For Au, (EIE = $-E_{LUMO}$ − (Au) has been predicted as **ascorbic acid** (3.94 eV = −1.16− (−5.10)) > **gallic acid** (3.47 eV = −1.63− (−5.10)) > **indomethacin** (2.94 eV = -2.16− (−5.10)) > $Z_1$ (2.35 eV = −2.75− (−5.10)) > $Z_2$ (2.16 eV = -2.94− (−5.10)) > $Z_3$ (1.52 eV = −3.58− (−5.10)). For Al, (EIE = $-E_{LUMO}$−(Al) has been estimated as **ascorbic acid** (2.92 eV = −1.16− (−4.08)) > **gallic acid** (2.45 eV = −1.63− (−4.08)) > **indomethacin** (1.92 eV = −2.16− (−4.08)) > $Z_1$ (1.33 eV = −2.75− (−4.08)) > $Z_2$ (1.14 eV = −2.94− (−4.08)) > $Z_3$ (0.5 eV = −3.58− (−4.08)).

The HIE were found to be as **ascorbic acid** (2.63 eV = −4.08−(-6.71) > $Z_3$ (2.49 eV = −4.08−(−6.57) > **gallic acid** (2.03 eV = −4.08−(−6.44) > **indomethacin** (1.92 eV = −4.08−(−6.00) > $Z_1$ (1.45 eV = −4.08−

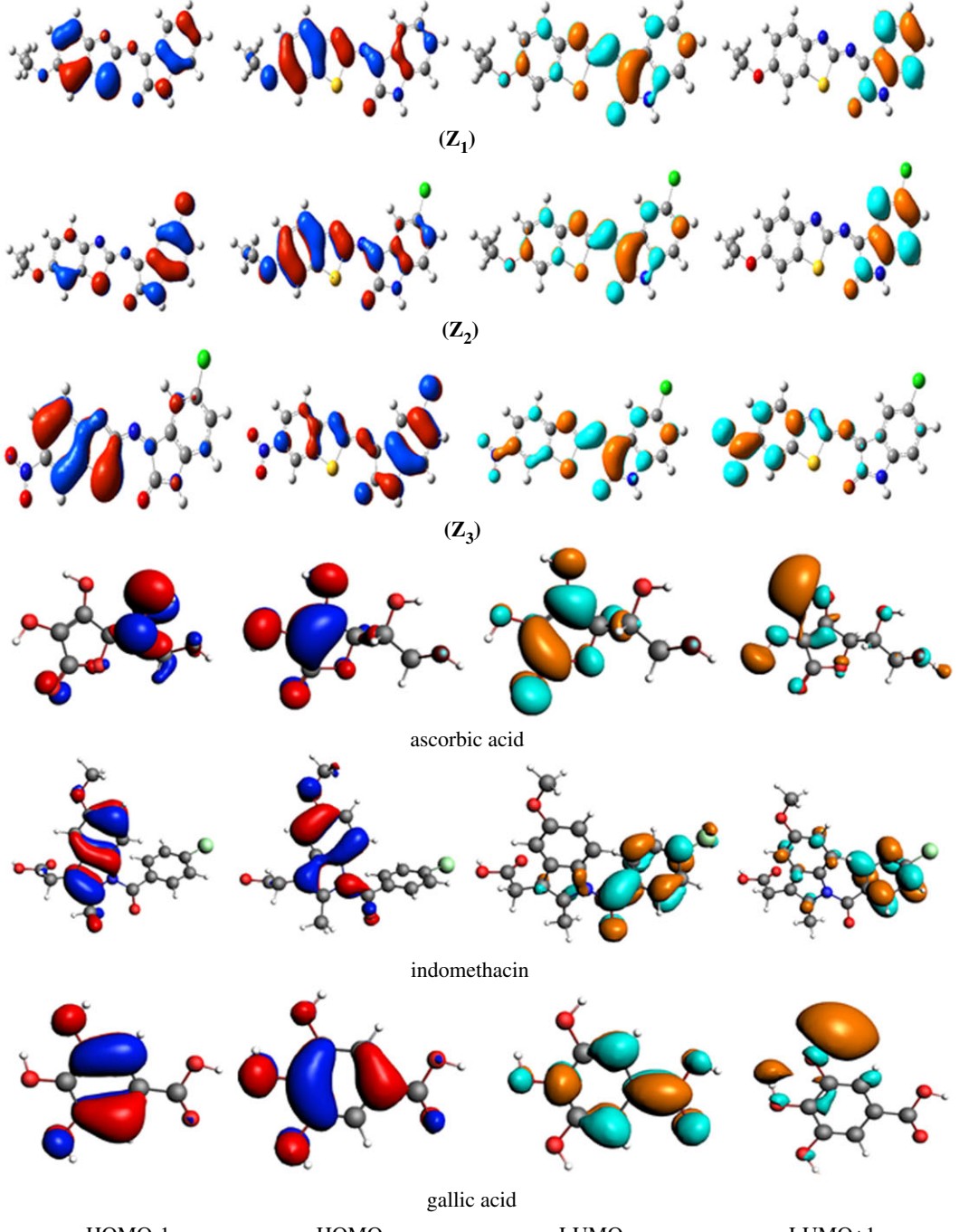

|          |      |      |         |
|----------|------|------|---------|
| HOMO−1   | HOMO | LUMO | LUMO+1  |

**Figure 2.** The charge density of FMOs of thiazole-derived ligands **(Z₁)**–**(Z₃)** and standard compounds at ground state (contour value = 0.035).

(−5.53) **> Z₂** (1.58 eV = −4.08−(−5.66) by taking Al electrode in consideration. For Au, HIE were found to be as **ascorbic acid** (1.61 eV = −5.10−(−6.71) > Z₃ (1.47 eV = −5.10−(−6.57) **> gallic acid** (1.34 eV = −5.10−(−6.44) > **indomethacin** (0.90 eV = −5.10−(−6.00) **> (Z₂)** (0.56 eV = −5.10− (−5.66) **> (Z₁)** (0.43 eV = −5.10−(−5.53). It can be seen that electron/hole injection energy barrier from thiazole ligand to electrode in ligands **(Z₃)/(Z₁)** is reduced in comparison with other compounds which is clarifying that in ligands **(Z₃)/(Z₁),** there would be better electron/hole injection ability than other counterparts.

The chemical systems with minor $E_{HOMO}$ values mostly exhibited fragile electron-donating aptitude. Table 1 demonstrates that both the ligands **(Z₁)** and **(Z₂)** might have enhanced electron-donating ability as compared with studied standard compounds. It can be concluded that both these thiazole-derived ligands have improved antioxidant as well as biological competency. It is also demonstrated that all

R. Soc. Open Sci. **8**: 210910

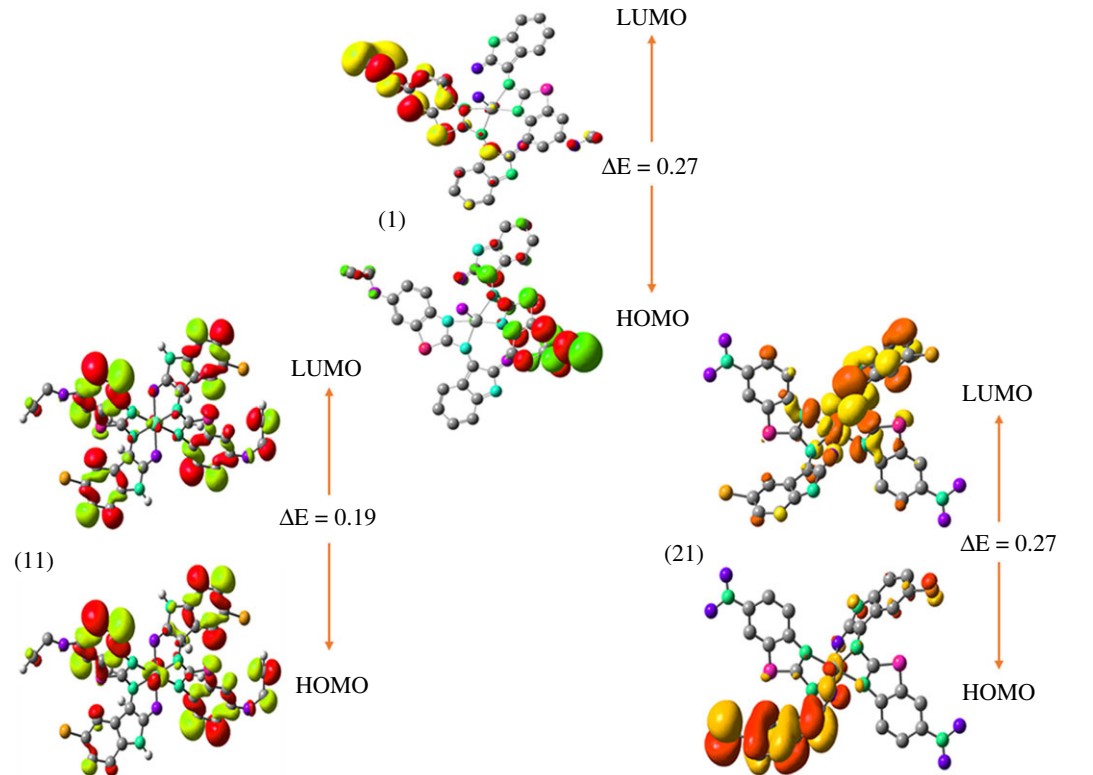

**Figure 3.** The charge density of FMOs of selected metal complexes at ground state (contour value = 0.042).

the ligands have a smaller energy gap of HOMO and LUMO ($\Delta E_{HOMO\ -LUMO}$) as compared with reference compounds. Among the three ligands, **(Z₃)** exhibited the highest energy difference as 2.99 eV, **(Z₁)** exhibited less energy difference as 2.78 eV, whereas **(Z₂)** exhibited the least value of energy gap as 2.72 eV. Overall, the order of energy difference is **(Z₃) > (Z₁) > (Z₂)**.

The energy gap difference of ligand **(Z₃)** was found to be considerably higher than the remaining two ligands **(Z₁)** and **(Z₂)** owing to the pronounced electron dragging ability of nitro ($NO_2$) group. The outcome of FMO study also revealed that the ligand **(Z₂)** exhibited a minimum energy gap contrary to other two ligands **(Z₁)** and **(Z₃)**, signifying the significant ICT communication within the ligand **(Z₂)**.

In indomethacin, the three-dimensional scattering of electronic charge density for LUMO+1 is only focused on the benzoyl group, LUMO at the chlorobenzoyl ring, whereas the HOMO and HOMO−1 at indole moiety. The ICT could be observed from HOMO and HOMO−1 of indole moiety to LUMO of chlorobenzoyl ring and LUMO+1 of benzoyl group. The ICT was experienced from HOMO−1 to LUMO+1, HOMO−1 to LUMO, HOMO to LUMO and HOMO to LUMO+1 in both types of standard compounds. In gallic acid, the electronic charge density of HOMO−1 was distributed on the benzene ring in addition to the oxygen of *m*-hydroxyl, the electronic charge density of HOMO was concentrated at the whole compound, the electronic charge density of LUMO was spread at carboxylic group as well as benzene ring, the electronic charge density of LUMO+1 was only focused at the hydroxyl (OH) groups.

The ICT could be established from HOMO and HOMO−1 of hydroxyl and benzene groups to LUMO of carboxylic group and LUMO+1 of hydroxyl groups. In the case of ascorbic acid, the ICT was also determined. In ligand **(Z₁)**, the electronic cloud of HOMO and HOMO−1 is mostly spread on the whole structure except ethoxy group, the charge density of LUMO is mostly scattered on the whole structure except ethoxybenzene part, whereas the charge density of LUMO+1 is distributed only on the isatin moiety. In ligand **(Z₂)**, the electronic charge density of LUMO and HOMO−1 is dispersed over the whole structure except ethoxybenzene part, and the charge density of HOMO is mostly distributed on the whole structure, whereas the charge density of LUMO+1 is distributed on the isatin moiety.

In ligand **(Z₃)**, the charge density of HOMO−1 is dispersed over the aminothiazole moiety, while the charge density of HOMO is distributed on the isatin moiety. While for LUMO, the charge density is randomly spread on some parts of the structure, while the charge density of LUMO+1 is dispersed on the structure of the ligand with the exception of chloroisatin moiety. In the background of selected metal-based compounds, the results demonstrated that the extent of electronic charge density of

HOMO was based on the π-charge of phenyl and methylene moieties, the lone pair of nitrogen atoms, as well as the M(II) ion of the $dx^2-y^2$ orbital. According to the above study, it was presumed that there is indeed a π-donation of electrons mainly establishing from thiazole ligands and contributing to a back-donation of both the metal d-orbital as it is from the transition metal d-orbital to the ligands, which together strengthens the metal complexes stabilization (figure 3).

## 3.10. Global reactivity descriptors

Global chemical reactivity descriptors (GCRD) are substantial characteristic features to understand and estimate the reactivity of biological active compounds, antioxidant ability and antibacterial aptitude of the drugs in addition to structure stability of any chemical system. These descriptors have also been stated as biological activity descriptors [58]. Here, we predicted various GCRD parameters of the studied thiazole ligands and their selected metal-based compounds including chemical hardness ($\eta$), chemical potential ($\mu$), electronegativity ($\chi$), global softness ($\sigma$) and electrophilicity index ($\omega$). The HOMO–LUMO energies have been used to calculate the important global reactivity parameters and the values are depicted in table 1, while the equations S1–S7 are given in the electronic supplementary material. The chemical hardness of any compound is interconnected with aromaticity. The chemical potential represents the affinity of electrons to move from the electronic cloud. The chemical hardness also signifies the degree of the hindrance of the electronic cloud to distort and electrophilicity index implies the stabilization energy of the studied compound when it is concentrated by the electrons from the external surrounding [59]. The values of reactivity descriptor and energy gap establish that the ligands sustain good reactivity. Any chemical structures having a small FMO energy gap (ΔE) are renowned to be less stable, more reactive, soft and vice versa [60].

In our studied ligands, the calculated values of global hardness have been obtained smaller than the values of global softness, which suggested greater reactivity and least stability of the studied ligands. The decreasing trend of FMO energy gaps and hardness values were found to be exactly similar. Furthermore, the values of chemical potential have also been used to elucidate the reactivity and stability of the studied ligands. The chemical systems with greater values of chemical potential are renowned as less reactive and more stable and vice versa [61]. Among studied ligands, **(Z₃)** exhibited the highest value of IP as 6.57 eV, whereas **(Z₁)** exhibited the least value of IP as 5.53 eV. The decreasing order of IP was **(Z₃) > (Z₂) > (Z₁)**. Ligand **(Z₃)** and **(Z₁)** possessed maximal value of EA as 3.58 eV and minimal value of EA as 2.7 eV, correspondingly. Consequently, the electron gaining as well as donating ability of studied ligands have been defined and explained by the values of EA and IP, respectively, which has a correlation with the energies of HOMO and LUMO orbitals.

In general, the values of IP were determined higher in magnitude in contrast with EA, which indicated that the studied ligands contain remarkable electron-donating capability which also supported the outcomes of global electrophilicity index ($\omega$). Among the investigated ligands, **(Z₃)** possessed the highest value of electronegativity as 5.07 eV, whereas **(Z₁)** possessed least value of electronegativity as 4.14 eV. The decreasing order of global electrophilicity ($\omega$) of the investigated ligands is as **(Z₃) > (Z₂) > (Z₁)**. From the obtained data, it has been concluded that all the three ligands **(Z₁)–(Z₃)** are competent chemically hard chemical systems having paramount kinetic stability as well as affective electron-donating ability.

## 3.11. One-electron transfer mechanism

When the free radical gains an electron from the antioxidant compounds, then the produced radical cation should be sufficiently stable for improved radical scavenging capability in one-electron transfer mechanism. Like this, the IP could be used for evaluating the antioxidant aptitude of investigated compounds. This is the physical characteristic factor, calculated as $IP = -E_{HOMO}$ and used to describe the range of electronic charge transfer [62]. It is also predicted that the radical scavenging action may be more for those studied compounds which exhibit the reduced value of IP. These results concluded that **(Z₁)** and **(Z₂)** ligands would have significant antioxidant potential that sounds auspicious to determine radical scavenging proficiency.

## 3.12. Molecular electrostatic potential analysis

The physical as well as chemical features of any chemical structure can be studied by means of MEP plots. Generally, MEP plots can be employed to understand the probable attacks of nucleophiles or electrophiles at

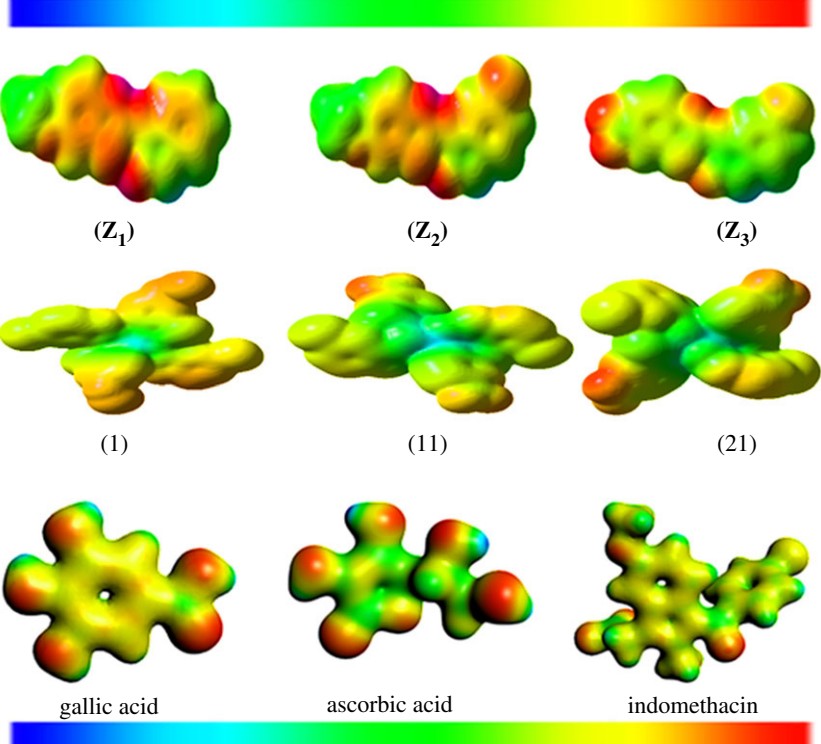

**Figure 4.** MEP surfaces of ligands **(Z₁)**–**(Z₃)** and their selected metal complexes and reference compounds.

the more appropriate sites in the chemical systems. MEP surfaces contain some characteristic colours such as green, orange, blue, red and yellow which demonstrated the magnitudes of electrostatic potential within the chemical systems. The ascending order of electrostatic potential extent was found to be: red > orange > yellow > green > blue. The red highlighted part on MEP plot was offered by the oxygen atoms and shows the area of negative potential and could be the most appropriate site for the attack of electrophiles. By contrast, blue or green highlighted part shows the area of positive potential and could be the most appropriate site for the attack of nucleophiles. The blue and green colours were typically shown for hydrogen atoms as well as some carbon atoms which constitute electron-deficient zones [63]. In figure 4, the MEP plots for ligands and metal complexes in addition to reference compounds have been illustrated.

It could be observed from the figure that in the MEP plot of gallic acid, the negative and positive electrostatic potentials were concentrated on the oxygen and all hydrogen atoms of hydroxyl and carboxylic groups, individually. In indomethacin, the negative electrostatic potential was spread on the oxygen atoms of carbonyl, carboxylic as well as methoxy groups; however, the positive electrostatic potential was distributed on the hydrogen atoms of carboxylic and methoxy groups, similarly for the ascorbic acid. In the current study, the negative electrostatic potential could be observed on the oxygen atoms of ligands, while the positive electrostatic potential is concerted on the hydrogen atom of –NH groups present in the thiazole ligands.

## 3.13. Natural bond orbital analysis

This approach is based on the premise that it is used to study the intra- and inter-molecular interaction and similarities between different bonds. This analysis is an excellent tool for examining the hyper-conjugative engagement and electron transfer phenomena from the loaded lone pair of electrons for chemical interpretation. For this purpose, the stabilizing energy ($E^2$) of the delocalized donor ($i$) is calculated by applying equation (3.1) to the acceptor ($j$) orbitals.

$$E^2 = q_i \frac{F_{ij}^2}{\varepsilon_j - \varepsilon_i}. \tag{3.1}$$

The stabilization energy $E^2$ has a direct relation with the NBO intensity interaction. This also provides a simple framework to examine the interfaces in either empty or occupied orbital spaces, in addition to

charging transformation and conjugation encounter in the chemical structure [64]. The calculations of NBO analysis for ligands $(Z_1)$–$(Z_3)$ were conducted using the DFT/B3LYP/6-31 + G(d,p) strategy. This is an important practice for finding the conjugation or electronic charge transfer in specific environments of the chemical system with more accurate results. The sum of the stabilization energy is strongly related to the frequency of the hyper-conjugative interaction between the electron receptor and electron donor orbitals. Likewise, the conjugation of the complete chemical structure can be described by using second-order stabilization capability. The strong association between the contributors and acceptors resulted in a huge amount of stabilization energy. The NBO results of the ligands are summarized in the electronic supplementary material, tables S14–S16. The representative transitions for ligands $(Z_1)$–$(Z_3)$ are discussed herein. Various electronic transitions could be observed as $\pi(C_{18}\text{-}C_{19}) \rightarrow \pi^*(C_{14}\text{-}C_{15})$, $\pi(C_{16}\text{-}C_{17}) \rightarrow \pi^*(C_{14}\text{-}C_{15})$, $\pi(C_8\text{-}C_9) \rightarrow \pi^*(C_6\text{-}C_7)$, $\pi(C_{14}\text{-}C_{15}) \rightarrow \pi^*(C_{12}\text{-}N_{13})$ and $\pi(C_2\text{-}O_3) \rightarrow \pi^*(C_4\text{-}N_{23})$ with approximations of stabilization energy as 11.85, 9.30, 5.89, 5.15 and 4.89 kcal mol$^{-1}$, correspondingly for thiazole ligand $(Z_1)$.

Likewise the electronic transitions: $\pi(C_5\text{-}C_6) \rightarrow \pi^*(C_4\text{-}N_{23})$, $\pi(C_7\text{-}C_8) \rightarrow \pi^*(C_5\text{-}C_6)$, $\pi(C_{18}\text{-}C_{19}) \rightarrow \pi^*(C_{14}\text{-}C_{15})$, $\pi(C_{16}\text{-}C_{17}) \rightarrow \pi^*(C_{14}\text{-}C_{15})$, and $\pi(C_9\text{-}C_{10}) \rightarrow \pi^*(C_5\text{-}C_6)$ were obtained with stabilization energy values of 15.69, 14.84, 12.01, 9.29 and 5.76 kcal mol$^{-1}$, correspondingly for ligand $(Z_2)$. Furthermore, the transitions including $\pi(C_{16}\text{-}C_{17}) \rightarrow \pi^*(N_{13}\text{-}C_{14})$, $\pi(C_{15}\text{-}C_{19}) \rightarrow \pi^*(N_{13}\text{-}C_{14})$, $\pi(C_7\text{-}C_8) \rightarrow \pi^*(C_5\text{-}C_6)$, $\pi(C_5\text{-}C_6) \rightarrow \pi^*(C_7\text{-}C_8)$, $\pi(N_{13}\text{-}C_{14}) \rightarrow \pi^*(C_{12}\text{-}N_{21})$ and $\pi(C_9\text{-}C_{10}) \rightarrow \pi^*(C_5\text{-}C_6)$ have been found with stabilization energies like 29.48, 24.58, 20.77, 19.27, 19.17 and 15.28 kcal mol$^{-1}$, correspondingly for thiazole ligand $(Z_3)$. The most probable electronic transitions were observed as $LP(C_5) \rightarrow \pi^*(C_4\text{-}N_{23})$, $LP(N_1) \rightarrow \pi^*(C_2\text{-}O_3)$ and $LP(O_3) \rightarrow \pi^*(N_1\text{-}C_2)$ with stabilization energy values as 47.01, 21.28, 127.00 kcal mol$^{-1}$ for ligands $(Z_1)$, $(Z_2)$ and $(Z_3)$, respectively. The dynamic values of stabilization energy concluded that the studied ligands have been promised with the hyper-conjugative interactions, ICT in addition to extended conjugation that is the most important explanation for the stability of synthesized ligands.

## 3.14. Mulliken atomic charge analysis

In quantum chemical study, MAC analysis has played a vital role to explore the molecular electronic structure, dipole moment, molecular polarizability and electronic properties of any chemical structure [65]. The MACs of ligands were calculated by employing B3LYP level with a 6-31 + G(d,p) basis set. Ligands possessed different types of atoms (C, O, N, H, S, Cl) and exhibited a definite positive value of Mulliken charges based on their electronegativity. Nitrogen and oxygen atoms of ligands possessed negative charges, whereas all the hydrogen atoms showed positive charges. The MACs are depicted in the electronic supplementary material, table S17, while the histogram given in figure 5 is the better demonstration of the MAC distribution.

The calculated MACs have shown the occurrence of more electronegative atoms such as (1N = −0.847978), (3O = −0.495645), (13N = −0.803953), (20O = −0.554744) and (23N = −0.754526) for ligand $(Z_1)$, (1N = −0.849475), (3O = −0.490926), (13N = −0.802164), (20O = −0.554104) and (23N = −0.752910) for ligand $(Z_2)$, (1N = −0.849787), (3O = −0.497887), (13N = −0.793382), (20N = −0.144445), (21N = −0.740957), (23O = −0.384794) and (24O = −0.335241) for ligand $(Z_3)$. Additionally, it is observed that the carbon atoms C1 and C18 comprise positive charges as they have direct interaction with oxygen atoms, which are more electronegative than carbon. Likewise, C4, C10, C12 and C14 also carry positive charges as they are directly attached with nitrogen atoms, which are more electronegative than carbon. Whereas all the hydrogen atoms contain positive charges owing to the attached comparatively negatively charged carbon atoms.

## 3.15. FT-IR analysis

The number of atoms with either a symmetry point group or efficient vibratory normal modes was found to be the same in the experimental FT-IR. The FT-IR modes of carbonyl groups were located at 1706, 1708 and 1768 cm$^{-1}$ for ligands $(Z_1)$, $(Z_2)$ and $(Z_3)$, respectively (electronic supplementary material, figure S26, tables S18–S20). The vibrational frequency for N-H group was obtained in the range 3510–3600 cm$^{-1}$ for all the ligands. The C-H vibrations of hetero atomic organic compounds along with their derived compounds are very similar to C-H vibrations in the benzene system. They contain several small bands that are due to the stretching vibration of C-H bond. The stretch frequencies of aromatic ring C-H for the current study are seen at a frequency of 3195–3411 cm$^{-1}$ in the FT-IR spectra. Analysis for all such excitation energies has been developed through the experimental FT-IR spectra. In this analysis, a strong C=N vibration pattern was also observed at 1626–1768 cm$^{-1}$.

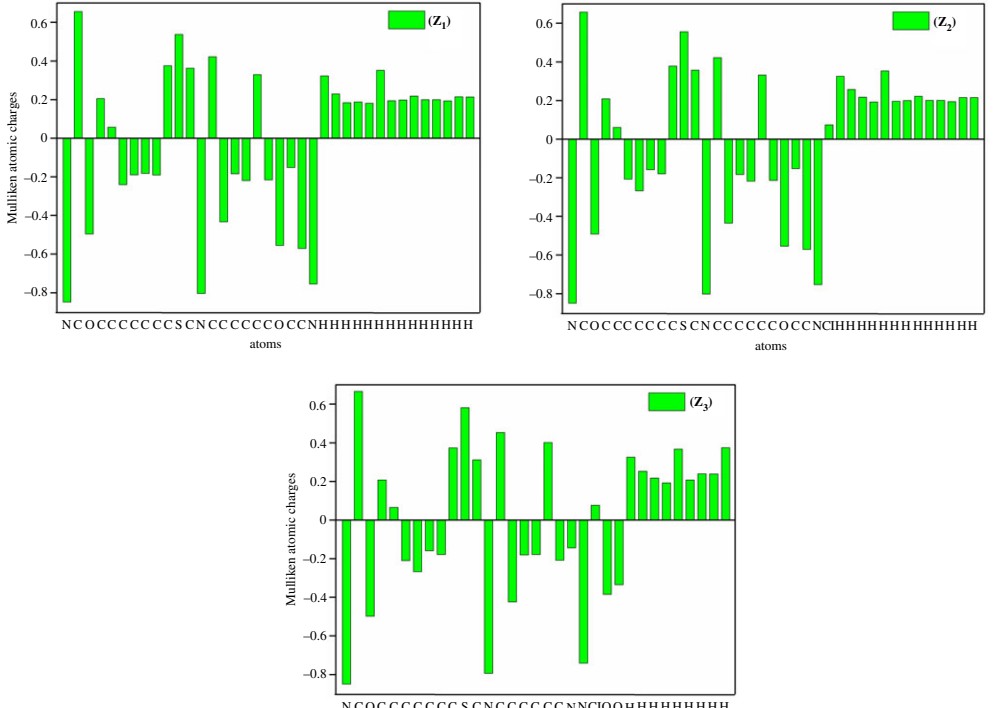

**Figure 5.** MAC analysis for thiazole ligands **(Z$_1$)**–**(Z$_3$)**.

## 3.16. UV-Vis analysis

The UV-Vis analysis of the studied ligands was carried out at their optimized geometries with the CAM-B3LYP feature as a part of the TD-DFT measurements, at the same basis set (6-31G(d,p) (electronic supplementary material, table S21 and figure S27). The computed UV-Vis spectra were reflecting the absorbance, the excitation energies and oscillator strengths for the studied ligands. For ligand **(Z$_1$)**, the first absorbance band was assessed at 628 nm having oscillator frequency $f^{1-4} = 0.0719$ with the main involvement of H → L (89%) and H−1 → L (6%) transitions. The second absorbance band transformation found at 556 nm having oscillator frequency $f^{1-4} = 0.0002$ was accredited to the major participation of H−2 → L (95%). Whereas the third absorbance band was obtained at 483 nm having oscillator frequency $f^{1-4} = 0.0888$ with the main contribution of H−1 → L (84%), H → L (4%), H → L + 1 (2%) and H → L (5%) transitions.

Similarly for ligand **(Z$_2$)**, the first absorbance band was observed at 642 nm having oscillator frequency $f^{1-4} = 0.0794$ with the main influence of H → L (89%) and H-1 → L (6%) transitions. The second absorbance band transformation found at 559 nm having oscillator frequency $f^{1-4} = 0.0001$ was accredited to the major participation of H-2 → L (96%). While the third absorbance band was assessed at 496 nm having oscillator frequency $f^{1-4} = 0.0971$ with the main participation of H-1 → L (86%), H → L (4%) and H → L (5%) transitions. Likewise, for ligand **(Z$_3$)**, the first absorbance band was observed at 674 nm having oscillator frequency $f^{1-4} = 0.7607$ with the main involvement of H → L (110%) and H−1 → L (6%) transitions. The next absorbance band was found at 580 nm having oscillator frequency $f^{1-4} = 0.0467$ due to the main participation of H−1 → L (92%) and H → L (8%) transitions. Whereas the third absorbance band transformation found at 489 nm having oscillator frequency $f^{1-4} = 0.0002$ was attributed to the major involvement of H−2 → L (99%). In comparison, the test results only showed two bands, an intensive, wide band centred on approximately 306–392 nm and another 253–298 nm in their experimental results.

## 3.17. Antibacterial activity

Schiff base ligands **(Z$_1$)**–**(Z$_3$)** in addition to their derived transition metal chelates were subjected to antibacterial screening against two Gram (−) bacteria including *Escherichia coli* and *Salmonella typhimurium* as well as two Gram (+) bacteria including *Staphylococcus aureus* and *Bacillus subtilis*. Inhibition zones of compounds against Gram (+) and Gram (−) bacterial strains are illustrated in the

**Table 2.** Antibacterial and antioxidant activities of ligands and their derived metal complexes. $SD^1$ = Streptomycin, $SD^2$ = BHT.

| comp. | antibacterial activity, zones of inhibition (mm) | | | | antioxidant activity | |
|---|---|---|---|---|---|---|
| | B. subtilis | S. aureus | E. coli | S. typhimurium | % inhibition (1 mg ml$^{-1}$) | % inhibition (2 mg ml$^{-1}$) |
| (Z$_1$) | 04 | 10 | 10 | 12 | 30.11 | 14.80 |
| (Z$_2$) | 15 | 22 | 26 | 25 | 33.33 | 10.70 |
| (Z$_3$) | 22 | 22 | 19 | 13 | 36.83 | 15.05 |
| (1) | 10 | 08 | 09 | 07 | 93.18 | 36.02 |
| (2) | 05 | 12 | 17 | 02 | 54.05 | 21.13 |
| (3) | 10 | 18 | 05 | 04 | 68.01 | 29.21 |
| (4) | 09 | 05 | 09 | 09 | 58.13 | 12.34 |
| (5) | 12 | 05 | 10 | 11 | 76.06 | 46.42 |
| (6) | 09 | 10 | 12 | 13 | 89.42 | 38.09 |
| (7) | 15 | 21 | 20 | 25 | 59.59 | 25.06 |
| (8) | 20 | 21 | 22 | 24 | 68.01 | 34.23 |
| (9) | 20 | 25 | 27 | 27 | 47.87 | 21.41 |
| (10) | 20 | 19 | 25 | 25 | 73.73 | 38.23 |
| (11) | 20 | 30 | 29 | 27 | 24.82 | 08.03 |
| (12) | 15 | 19 | 25 | 16 | 70.11 | 41.27 |
| (13) | 16 | 26 | 19 | 25 | 77.44 | 45.15 |
| (14) | 20 | 28 | 25 | 25 | 40.11 | 13.19 |
| (15) | 20 | 20 | 23 | 23 | 42.90 | 26.27 |
| (16) | 22 | 20 | 20 | 20 | 26.83 | 11.36 |
| (17) | 20 | 22 | 26 | 25 | 74.05 | 40.65 |
| (18) | 21 | 26 | 18 | 22 | 08.36 | 03.04 |
| (19) | 04 | 19 | 12 | 19 | 29.94 | 12.23 |
| (20) | 12 | 10 | 11 | 06 | 44.36 | 25.56 |
| (21) | 20 | 23 | 20 | 24 | 23.36 | 12.32 |
| (SD$^1$) | 21 | 32 | 30 | 26 | — | — |
| (SD$^2$) | — | — | — | — | 74.80 | 54.30 |

electronic supplementary material, figure S28. The results of antibacterial actions of the synthesized compounds were assessed in comparison with the standard drug, streptomycin (as given in table 2 and figure 6). It inhibited 32, 30 26 and 21 mm zones of bacteria *Staphylococcus aureus*, *Escherichia coli*, *Salmonella typhimurium* and *Bacillus subtilis*, respectively. From the synthesized ligands, (Z$_2$) was found to be the most active ligand and showed maximum activity of 26, 25 and 22 mm against *Escherichia coli*, *Salmonella typhimurium* and *Staphylococcus aureus*, respectively, while (Z$_3$) was moderately active and showed maximum activity of 22 mm against two bacterial strains, *Bacillus subtilis* and *Staphylococcus aureus*. And (Z$_1$) was the least active among the ligands with the lowest activity of 4 mm against *Bacillus subtilis*.

Among the derived transition metal complexes, (11) exhibited the highest activity of 30, 29 and 27 mm activity against *Staphylococcus aureus*, *Escherichia coli* and *Salmonella typhimurium*, correspondingly and moderate activity of 20 mm against *Bacillus subtilis*. Complex (9) showed the highest activity of 27 mm against two bacterial strains, *Escherichia coli* and *Salmonella typhimurium*, and moderate activity of 25 and 20 mm for *Bacillus subtilis* and *Staphylococcus aureus*, individually. Similarly, complex (14) was also found to be more active with an inhibition zone of 28, 25, 25 and 20 mm for *Staphylococcus aureus*, *Escherichia coli*, *Salmonella typhimurium* and *Bacillus subtilis*, respectively. Complexes (7), (10), (13) and (17) exhibited moderate activity and inhibited 25 mm zone area of *Salmonella typhimurium*. Complex (16) showed

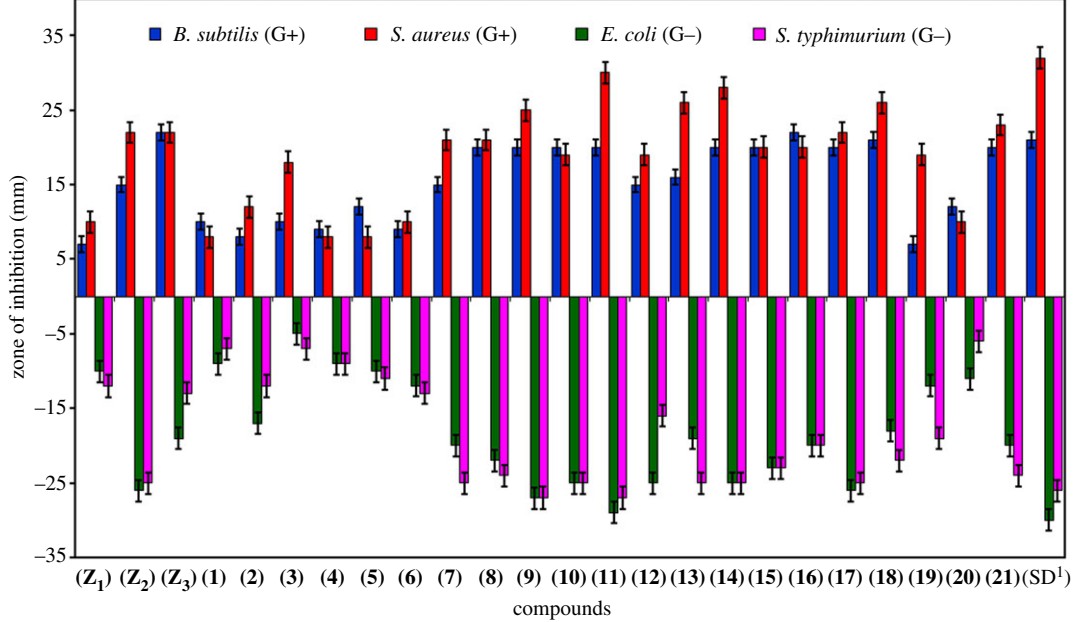

**Figure 6.** Antibacterial activity of aminothiazole ligands (Z₁)–(Z₃) versus their derived transition metal complexes (1)–(21).

maximum activity with 22 mm inhibition, complex **(18)** showed moderate activity with 21 mm inhibition, whereas complex **(19)** showed the least activity with 4 mm inhibition against *Bacillus subtilis*.

However, complex **(3)** was the least active with 5 and 4 mm inhibitory activity against *Escherichia coli* and *Salmonella typhimurium*, individually. Complex **(2)** also showed minimum activity of 5 and 2 mm inhibition against *Bacillus subtilis* and *Salmonella typhimurium*, respectively. Both complexes **(4)** and **(5)** showed the least activity of 5 mm against *Staphylococcus aureus*. The obtained antibacterial results demonstrated that most of the scrutinized compounds have shown moderate to significant antibacterial efficacy in contrast with streptomycin (standard drug). But overall, the antibacterial activity of transition metal complexes is higher than that of their corresponding uncomplexed ligands.

## 3.18. Antioxidant activity

The DPPH is stable free radical and was used to determine the radical scavenging aptitude of all the thiazole ligands along with their derived transition metal complexes by using samples at two concentration levels (1 and 2 mg ml⁻¹). The antioxidant potential of the synthesized compounds was determined by comparing their results with that of standard drug, BHT as given in table 2 and figure 7. The results presented that among 1 mg ml⁻¹ sample concentrations, oxovanadium complex **(1)** has shown highest (93.18%), while the cobalt complex **(18)** exhibited least (08.36%) antioxidant activity. Likewise for 2 mg ml⁻¹ sample concentrations, nickel complex **(5)** showed maximum (46.42%), whereas cobalt complex **(18)** exhibited minimum (03.04%) antioxidant activity. Overall, the cobalt complex **(18)** was concluded as least active compound as it exhibited minimum activity in both sample concentration.

# 4. Conclusion

The medically important metal-based thiazole-derived Schiff base compounds have been designed, synthesized and experimentally and theoretically characterized. The spectral analysis confirmed the synthetic mode of all the compounds by using the oxygen of isatin moiety, nitrogen of azomethine linkage and nitrogen of thiazole moiety. Based on spectral and magnetic study results, octahedral geometries were recommended for all the complexes except for oxovanadium complexes **(1), (8) and (15)** which exhibited square-pyramidal geometry. The molar conductance results verified the electrolytic nature of all the metal chelates. Quantum chemical calculations based on DFT study have been satisfactorily used to obtain the optimized molecular structures and electronic calculations of aminothiazole ligands at B3LYP/6-31 + G(d,p) level of theory. The calculated values of global

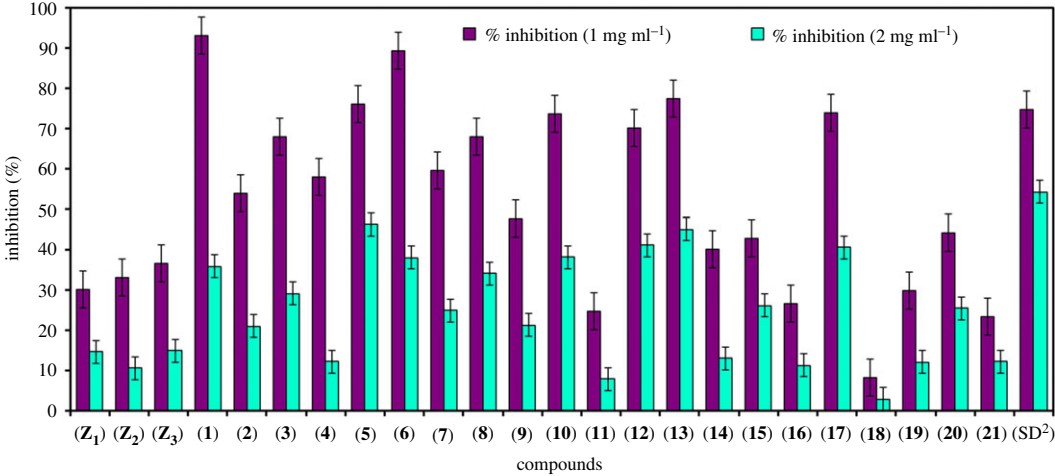

**Figure 7.** Antioxidant activity of aminothiazole ligands **(Z₁)**–**(Z₃)** versus their derived transition metal complexes **(1)**–**(21)**.

hardness have been obtained smaller than the values of global softness, which suggested less stability and high chemical reactivity of the studied ligands. The decreasing trend of FMO energy gaps and softness values were found to be exactly similar to **(Z₃) > (Z₁) > (Z₂)**. The FMO study, electrostatic potential capacity, molecular descriptors and MACs signified that these thiazole-derived ligands are active natural bioactive contenders. All the synthesized compounds have also been subjected to antibacterial and antioxidant screening. All the compounds showed comparable results of antibacterial and antioxidant activity with that of standard drugs but complexes of Zn(II), Co(II) and VO(IV) have shown significant antibacterial activity and Ni(II), Cu(II) and VO(IV) complexes exhibited more pronounced antioxidant properties. The reason for the enormous results is the heterocyclic ring containing the N and S heteroatoms (thiazoles). Overall, metal complexes exhibited higher biological activities as compared with their respective uncomplexed ligands owing to chelation. The bio-activity data have evidently displayed that all these thiazole-derived synthetic compounds were potent antimicrobial agents and could be helpful for the pharmaceutical industry to reduce or inhibit the growth of the pathogens.

Data accessibility. Data (IR and computational) are provided in the electronic supplementary material, file where applicable.
Authors' contributions. S.H.S. was involved in conceptualization, funding acquisition, project administration, supervision and writing—review and editing. Z.A. was involved in writing—original draft. W.Z. was involved in data curation, resources and validation. M.A. was involved in data curation. A.U.H. was involved in DFT software. E.U.M. was involved in technical evaluation. A.I. was involved in software. M.I. was involved in formal analysis.
Competing interests. The authors report no declarations of interest.
Funding. The authors thank the Higher Education Commission (HEC), Islamabad, Pakistan for financial assistance through NRPU Project no. 7800. Authors extend their appreciation to the Deanship of Scientific Research at King Khalid University for funding this work through research groups programme under grant no. R.G.P.2/24/42.

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
