## [Peer Review File · Royal Society Open Science]

Review History

RSOS-210910.R0 (Original submission)

Review form: Reviewer 1 (Akhtar Hussain)

Is the manuscript scientifically sound in its present form?

No

Are the interpretations and conclusions justified by the results?

Yes

Is the language acceptable?

No

Do you have any ethical concerns with this paper?

No

Have you any concerns about statistical analyses in this paper?

Yes

Recommendation?

Major revision is needed (please make suggestions in comments)

Comments to the Author(s)

In this paper, Sumrra and co-workers report on the development of a series of homoleptic Schiff base complexes of first-row transition metals, namely, VO(IV), Cr(III), Fe(II), Co(II), Ni(II), Cu(II), and Zn(II). Theoretical DFT studies performed at a suitable level of theory have also been reported. The Schiff base ligands and the corresponding complexes were synthesized in satisfactory yields and characterized by routine methods. DFT studies were conducted to corroborate the experimental characterization data. The authors then went on to study the antibacterial efficacy of the compounds and demonstrated that the ligands and the metal chelates were significantly active against both gram (+) and gram (-) cultured strains. The activities were referenced to the standard antibiotic streptomycin. Similarly, several complexes were demonstrated to have antioxidant activity. The biological activities of the chelates were found to be higher compared to the ligands alone.

Medicinal inorganic chemistry is a growing area of research and there is great scope to explore the potential of metal complexes for therapeutic applications. Similarly, antibacterial studies of new molecules with a novel mechanism of action are of importance considering the emergence of drug-resistant bacteria and AMR.

After a careful review of the manuscript, I find that there are some important issues to address before I can recommend this work for publication. Addressing these issues will significantly improve the overall quality and the scientific soundness of the manuscript.

[1] My greatest concern is regarding the molar conductivity measurements in DMF. The conductivity values should be dependent on the electrolytic nature of the compounds and independent of the counteranion. The trend in conductivity should be $1:3$ or $3:1 > 1:2$ or $2:1 > 1:1$. See for example, W. J. Geary, *Coord. Chem. Rev.*, 7, 81 (1971). Thus, the trend observed for the complexes is not in line with the principle. Therefore, I suspect that the complexes might contain ionic impurities. The authors have not made any attempt to purify the complexes after synthesis.

[2] What is the likely mechanism of antibacterial and antioxidant activity of the complexes? What is the role of the central metal ion in the overall activity?

[3] The antibacterial activity should be quantified by determining the minimum inhibitory concentration (MIC) of the complexes. Besides, studies using cultured bacterial strains are now regarded as preliminary and do not provide a realistic assessment of the antibacterial potency. Bacterial biofilm formation is the main culprit for the development of antibiotic resistance and the severity of most infections that are difficult to treat (*Nature Rev. Microbiol.*, 14, 563 (2016)). So, the compounds should be tested for their ability to inhibit bacterial biofilm formation on model surfaces.

[4] The antibacterial activity is normally due to the generation of ROS, while the antioxidant activity could be due to the lowering of ROS concentration by the compounds. Thus, a compound with a greater ability to generate ROS would be more desirable as an antibacterial agent, while the one with greater ability to lower the concentration of ROS would be more desirable as an antioxidant. Thus, a compound showing both antibacterial and antioxidant activity seems to have a contradictory effect. The authors need to explain this fact.

[5] Why did the authors consider Cr(III) as the trivalent metal for their study? In my opinion, Fe(III) and Co(III) would have been more justified considering their bioessential nature. Cr(III) is not a good choice due to the fact that it can generate potentially carcinogenic other Cr species by speciation.

[6] Establishing the true identity and stability of a metal complex in a solution phase (preferably aqueous buffer such as PBS) is very important for any biological applications. The authors have not made any attempt to study the stability and solution identity of the reported complexes. Elemental analysis data do not provide any indication of the solution stability of a

complex. The stability of a complex (in DMF/PBS) can be qualitatively investigated by UV-visible spectroscopy by monitoring any spectral changes over a reasonable duration (say, up to 48 h).

[7] The authors have presented the MS data only for the ligands. Why not for the complexes? The measurement of ESI-MS for the complexes and the observation of a prominent peak corresponding to the complex would also tell whether the complex is stable in a solution phase or not.

[8] I have concerns regarding the stability of the VO(IV) complexes 1, 8, and 15. The two ligands are shown to bind the oxovanadium moiety as bidentate ligands. But, the ligand is forming a 3-membered ring. In coordination chemistry, 3-membered rings are generally strained and lead to the formation of unstable chelates unless stabilized by strong and special electronic factors.

[9] Did the authors try to obtain the single-crystal X-ray diffraction structures of the complexes? It is a powerful and valuable tool in coordination chemistry. I strongly suggest that the authors should always try to report crystal structures (at least one or two if not all) in their future manuscripts.

[10] NMR data of the ligands: please provide the coupling constant (J) values in the data section.

[11] Please provide the DFT coordinates of the optimized structures in the supporting information. Besides, the HOMO/LUMOs of the remaining compounds should be provided as supporting information.

[12] Antibacterial study: The zone-inhibition images for all the compounds should be provided as supporting information.

[13] Figures 6 & 7: The error bars are really missing and should be added.

[14] The graphical abstract looks are clumsy to me. The authors could present a simplified one by considering only three aspects, namely, DFT, antibacterial, and antioxidant studies. Similarly, the title of the manuscript could be improved.

[15] I suggest that the authors go for one round of improvement of the English used in the manuscript in order to improve the readability

Review form: Reviewer 2

Is the manuscript scientifically sound in its present form?

No

Are the interpretations and conclusions justified by the results?

No

Is the language acceptable?

Yes

Do you have any ethical concerns with this paper?

No

Have you any concerns about statistical analyses in this paper?

No

Recommendation?

Major revision is needed (please make suggestions in comments)

Comments to the Author(s)

The synthesis of Schiff bases from the condensation of 2-amino-6-ethoxybenzothiazole or 2-amino-6-nitrobenzothiazole with 5-chloroisatin or isatin and their 3d-transition metals complexes is reported in this ms. The compounds were characterized by spectroscopic methods (UV-Vis, FT-IR, ¹H-NMR, ¹³CNMR), mass spectrometry, elemental analysis, and by computational calculations employing the B3LYP/6-31+G(d,p) functional of DFT. The in vitro antibacterial potential of the compounds is also reported.

The manuscript can be published in Royal Society Open Science only after major revision and reconsideration.

The crystal structures would be supportive for the characterization of the 24 compounds. Was any crystallization attempted?

The structural and biological study of the Schiff base derived from 2-Amino-6-methoxybenzothiazole with Isatin and its metal complexes has been already published by Hassan et.al. (2020 International Journal of Pharmaceutical Research 12(1), pp. 323-336). This work should be taken into consideration during the preparations of this manuscript.

The Schiff base of 5-chloro isatin with 2-amino 5-nitrobenzothiazole and its complexes with ZnCl₂ and CuCl₂ are already reported by Shakir M et.al in Journal of Photochemistry and Photobiology B: Biology Volume 157, Pages 39 – 56, 2016. Please rationalize the repetition of their quoting here. Similarly, the spectroscopic data have already reported.

Please check the electric charges balance of the formulae shown in Scheme 1. E.g Cr(III) needs 3 CH₃COO⁻ anions and not one as it is shown. There are two Cl⁻, counter anions in the case of Co(II). Similarly Co(II), Cu(II) and Ni(II) dichloride

Please keep the chemical nomenclature. E.g

In IR there are “vibrational” bands

NMR resonance signals not “peaks”, “peaks for the methylene”, “carbon C4 peak” etc there are molecular fragments or ions in Mass spectrometry and not “peaks” etc

Infra red and NMR Spectra

A table with the characteristic vibrational bands and NMR signals of the compounds should be added in the text.

Molar conductance

Please check the consistence of the Molar conductance values (in table S2) and the formula of the compounds given in table S1. E.g. DMF solutions of electrolyte with 2 ions exhibit molar conductance value 65-90, while the corresponding value for the solution of electrolyte with 3 ions is 130-170 Ω⁻¹cm²mol⁻¹. The values measured correspond to 2 ions electrolydes which is not the case of proposed formulae in the case of Cr(III) acetate, copper dichloride, nickel dichloride cobalt dichloride etc.

Antibacterial activity

Please give the diameter of the paper used in the inhibition zones (IZ). How the IZ diameter of e.g 4 mm in case of Z1 is measured?

Figure 6. The negative values in the histograms of the antibacterial activity of aminothiazole ligands (Z1)-(Z3) vs their transition metal complexes (1)-(21) is not clear. Please explain.

Decision letter (RSOS-210910.R0)

Dear Dr Sumrra:

Title: Synthesis of Computed Aminothiazole Based Ligands and Their Endowed Transition Metal Chelates: In-vitro Antibacterial and Antioxidant Evaluation
Manuscript ID: RSOS-210910

The editor assigned to your manuscript has now received comments from reviewers. We would like you to revise your paper in accordance with the referee and Subject Editor suggestions which can be found below (not including confidential reports to the Editor). Please note this decision does not guarantee eventual acceptance.

Please submit your revised paper before 11-Aug-2021. Please note that the revision deadline will expire at 00.00am on this date. If we do not hear from you within this time then it will be assumed that the paper has been withdrawn. In exceptional circumstances, extensions may be possible if agreed with the Editorial Office in advance. We do not allow multiple rounds of revision so we urge you to make every effort to fully address all of the comments at this stage. If deemed necessary by the Editors, your manuscript will be sent back to one or more of the original reviewers for assessment. If the original reviewers are not available we may invite new reviewers.

On behalf of the Subject Editor Professor Anthony Stace and the Associate Editor Dr Andrew Harned.

RSC Associate Editor:

Comments to the Author:

The reviewers have expressed some enthusiasm for this work, but have raised several valid concerns regarding the experimental details. In addition, Reviewer 2 has identified two existing publications that appear to detail similar results. Any revised manuscript should be prepared to avoid duplication of previously published results.

RSC Subject Editor:

Comments to the Author:

(There are no comments.)

Reviewers' Comments to Author:

Reviewer: 1

Comments to the Author(s)

In this paper, Sumrra and co-workers report on the development of a series of homoleptic Schiff base complexes of first-row transition metals, namely, VO(IV), Cr(III), Fe(II), Co(II), Ni(II), Cu(II), and Zn(II). Theoretical DFT studies performed at a suitable level of theory have also been reported. The Schiff base ligands and the corresponding complexes were synthesized in satisfactory yields and characterized by routine methods. DFT studies were conducted to corroborate the experimental characterization data. The authors then went on to study the antibacterial efficacy of the compounds and demonstrated that the ligands and the metal chelates were significantly active against both gram (+) and gram (-) cultured strains. The activities were referenced to the standard antibiotic streptomycin. Similarly, several complexes were demonstrated to have antioxidant activity. The biological activities of the chelates were found to be higher compared to the ligands alone.

Medicinal inorganic chemistry is a growing area of research and there is great scope to explore the potential of metal complexes for therapeutic applications. Similarly, antibacterial studies of new molecules with a novel mechanism of action are of importance considering the emergence of drug-resistant bacteria and AMR.

After a careful review of the manuscript, I find that there are some important issues to address before I can recommend this work for publication. Addressing these issues will significantly improve the overall quality and the scientific soundness of the manuscript.

[1] My greatest concern is regarding the molar conductivity measurements in DMF. The conductivity values should be dependent on the electrolytic nature of the compounds and independent of the counteranion. The trend in conductivity should be 1:3 or 3:1 > 1:2 or 2:1 > 1:1. See for example, W. J. Geary, *Coord. Chem. Rev.*, 7, 81 (1971). Thus, the trend observed for the complexes is not in line with the principle. Therefore, I suspect that the complexes might contain ionic impurities. The authors have not made any attempt to purify the complexes after synthesis.

[2] What is the likely mechanism of antibacterial and antioxidant activity of the complexes? What is the role of the central metal ion in the overall activity?

[3] The antibacterial activity should be quantified by determining the minimum inhibitory concentration (MIC) of the complexes. Besides, studies using cultured bacterial strains are now regarded as preliminary and do not provide a realistic assessment of the antibacterial potency. Bacterial biofilm formation is the main culprit for the development of antibiotic resistance and the severity of most infections that are difficult to treat (*Nature Rev. Microbiol.*, 14, 563 (2016)). So,

the compounds should be tested for their ability to inhibit bacterial biofilm formation on model surfaces.

[4] The antibacterial activity is normally due to the generation of ROS, while the antioxidant activity could be due to the lowering of ROS concentration by the compounds. Thus, a compound with a greater ability to generate ROS would more desirable as an antibacterial agent, while the one with greater ability to lower the concentration of ROS would be more desirable as an antioxidant. Thus, a compound showing both antibacterial and antioxidant activity seems to have a contradictory effect. The authors need to explain this fact.

[5] Why did the authors consider Cr(III) as the trivalent metal for their study? In my opinion, Fe(III) and Co(III) would have been more justified considering their bioessential nature. Cr(III) is not a good choice due to the fact that it can generate potentially carcinogenic other Cr species by speciation.

[6] Establishing the true identity and stability of a metal complex in a solution phase (preferably aqueous buffer such as PBS) is very important for any biological applications. The authors have not made any attempt to study the stability and solution identity of the reported complexes. Elemental analysis data do not provide any indication of the solution stability of a complex. The stability of a complex (in DMF/PBS) can be qualitatively investigated by UV-visible spectroscopy by monitoring any spectral changes over a reasonable duration (say, up to 48 h).

[7] The authors have presented the MS data only for the ligands. Why not for the complexes? The measurement of ESI-MS for the complexes and the observation of a prominent peak corresponding to the complex would also tell whether the complex is stable in a solution phase or not.

[8] I have concerns regarding the stability of the VO(IV) complexes 1, 8, and 15. The two ligands are shown to bind the oxovanadium moiety as bidentate ligands. But, the ligand is forming a 3-membered ring. In coordination chemistry, 3-membered rings are generally strained and lead to the formation of unstable chelates unless stabilized by strong and special electronic factors.

[9] Did the authors try to obtain the single-crystal X-ray diffraction structures of the complexes? It is a powerful and valuable tool in coordination chemistry. I strongly suggest that the authors should always try to report crystal structures (at least one or two if not all) in their future manuscripts.

[10] NMR data of the ligands: please provide the coupling constant (J) values in the data section.

[11] Please provide the DFT coordinates of the optimized structures in the supporting information. Besides, the HOMO/LUMOs of the remaining compounds should be provided as supporting information.

[12] Antibacterial study: The zone-inhibition images for all the compounds should be provided as supporting information.

[13] Figures 6 & 7: The error bars are really missing and should be added.

[14] The graphical abstract looks are clumsy to me. The authors could present a simplified one by considering only three aspects, namely, DFT, antibacterial, and antioxidant studies. Similarly, the title of the manuscript could be improved.

[15] I suggest that the authors go for one round of improvement of the English used in the manuscript in order to improve the readability

Reviewer: 2

Comments to the Author(s)

The synthesis of Schiff bases from the condensation of 2-amino-6-ethoxybenzothiazole or 2-amino-6-nitrobenzothiazole with 5-chloroisatin or isatin and their 3d-transition metals complexes is reported in this ms. The compounds were characterized by spectroscopic methods (UV-Vis, FT-IR, ¹H-NMR, ¹³C-NMR), mass spectrometry, elemental analysis, and by computational calculations employing the B3LYP/6-31+G(d,p) functional of DFT. The in vitro antibacterial potential of the compounds is also reported.

The manuscript can be published in Royal Society Open Science only after major revision and reconsideration.

The crystal structures would be supportive for the characterization of the 24 compounds. Was any crystallization attempted?

The structural and biological study of the Schiff base derived from 2-Amino-6-methoxybenzothiazole with Isatin and its metal complexes has been already published by Hassan et.al. (2020 International Journal of Pharmaceutical Research 12(1), pp. 323-336). This work should be taken into consideration during the preparations of this manuscript.

The Schiff base of 5-chloro isatin with 2-amino 5-nitrobenzothiazole and its complexes with $ZnCl_2$ and $CuCl_2$ are already reported by Shakir M et.al in Journal of Photochemistry and Photobiology B: Biology Volume 157, Pages 39 – 56, 2016. Please rationalize the repetition of their quoting here. Similarly, the spectroscopic data have already reported.

Please check the electric charges balance of the formulae shown in Scheme 1. E.g Cr(III) needs 3 CH_3COO^- anions and not one as it is shown. There are two Cl^- , counter anions in the case of Co(II). Similarly Co(II), Cu(II) and Ni(II) dichloride

Please keep the chemical nomenclature. E.g

In IR there are “vibrational” bands

NMR resonance signals not “peaks”, “peaks for the methylene”, “carbon C4 peak” etc

there are molecular fragments or ions in Mass spectrometry and not “peaks” etc

Infra red and NMR Spectra

A table with the characteristic vibrational bands and NMR signals of the compounds should be added in the text.

Molar conductance

Please check the consistence of the Molar conductance values (in table S2) and the formula of the compounds given in table S1. E.g. DMF solutions of electrolyte with 2 ions exhibit molar conductance value 65-90, while the corresponding value for the solution of electrolyte with 3 ions is 130-170 $\Omega^{-1}cm^2mol^{-1}$. The values measured correspond to 2 ions electrolydes which is not the case of proposed formulae in the case of Cr(III) acetate, copper dichloride, nickel dichloride cobalt dichloride etc.

Antibacterial activity

Please give the diameter of the paper used in the inhibition zones (IZ). How the IZ diameter of e.g 4 mm in case of Z1 is measured?

Figure 6. The negative values in the histograms of the antibacterial activity of aminothiazole ligands (Z1)-(Z3) vs their transition metal complexes (1)-(21) is not clear. Please explain.

Author's Response to Decision Letter for (RSOS-210910.R0)

See Appendix A.

Decision letter (RSOS-210910.R1)

Dear Dr Sumrra:

Title: Metal Incorporated Aminothiazole Derived Compounds: Synthesis, DFT Analysis, In-vitro Antibacterial and Antioxidant Evaluation
Manuscript ID: RSOS-210910.R1

It is a pleasure to accept your manuscript in its current form for publication in Royal Society Open Science. The chemistry content of Royal Society Open Science is published in collaboration with the Royal Society of Chemistry.

Yours sincerely,
Dr Ellis Wilde
Publishing Editor, Journals

On behalf of the Subject Editor Professor Anthony Stace and the Associate Editor Dr Andrew Harned.

RSC Associate Editor
Comments to the Author:

The authors have responded to all of the concerns raised by the previous referees to the best of their ability. Due to the global circumstances, there are a still a few requested experiments that are not possible to obtain. At this time, I do not think these experiments are critical to the overall conclusions drawn by the authors. As a result, I feel acceptance of this manuscript is warranted.

Reviewer(s)' Comments to Author:

Appendix A

Reply to Reviewers' Comments RSOS-210910

Title: Synthesis of Computed Aminothiazole Based Ligands and Their Endowed Transition Metal Chelates: *In-vitro* Antibacterial and Antioxidant Evaluation

Revised Title: Metal Incorporated Aminothiazole Derived Compounds: Synthesis, DFT Analysis, *In-vitro* Antibacterial and Antioxidant Evaluation

Editor Comments: The reviewers have expressed some enthusiasm for this work, but have raised several valid concerns regarding the experimental details. In addition, Reviewer 2 has identified two existing publications that appear to detail similar results. Any revised manuscript should be prepared to avoid duplication of previously published results.

Authors' reply: We appreciate your kindness for conveying the reviewer comments on our manuscript and overall facilitation regarding the reviewing process of the manuscript. We have clarified (in reply of 2nd comment) to Reviewer 2 and would also like to elaborate here that we have synthesized three aminothiazole Schiff base; (**Z₁**) from 2-amino-6-ethoxybenzothiazole and isatin, (**Z₂**) from 2-amino-6-ethoxybenzothiazole and 5-chloroisatin, and (**Z₃**) from 2-amino-6-nitrobenzothiazole and 5-chloroisatin. It can be clearly seen that we have synthesized new Schiff bases using aminothiazole moieties which are different from those used in the previously published recommended papers (2-amino-6-methoxybenzothiazole and 2-amino-5-nitrobenzothiazole). Therefore, there is no duplication of previously published results.

The envisioned manuscript has been edited according to the comments/suggestions of the referees. The point-wise responses to the reviewers' comments are given in BLUE in 'Reply to reviewers' comments, while the revision in the revised manuscript has been highlighted as YELLOW. We hope that the reviewers' comments are satisfactorily addressed and we look forward to see our article published in *Royal Society Open Science*.

Reviewer(s)' Comments to Author:

Reviewer: 1

In this paper, Sumrra and co-workers report on the development of a series of homoleptic Schiff base complexes of first-row transition metals, namely, VO(IV), Cr(III), Fe(II), Co(II), Ni(II), Cu(II), and Zn(II). Theoretical DFT studies performed at a suitable level of theory

have also been reported. The Schiff base ligands and the corresponding complexes were synthesized in satisfactory yields and characterized by routine methods. DFT studies were conducted to corroborate the experimental characterization data. The authors then went on to study the antibacterial efficacy of the compounds and demonstrated that the ligands and the metal chelates were significantly active against both gram (+) and gram (-) cultured strains. The activities were referenced to the standard antibiotic streptomycin. Similarly, several complexes were demonstrated to have antioxidant activity. The biological activities of the chelates were found to be higher compared to the ligands alone. Medicinal inorganic chemistry is a growing area of research and there is great scope to explore the potential of metal complexes for therapeutic applications. Similarly, antibacterial studies of new molecules with a novel mechanism of action are of importance considering the emergence of drug-resistant bacteria and AMR.

After a careful review of the manuscript, I find that there are some important issues to address before I can recommend this work for publication. Addressing these issues will significantly improve the overall quality and the scientific soundness of the manuscript.

Authors' reply: We would like to acknowledge the reviewer for the encouraging review of our manuscript, and for the comments, corrections and suggestions that ensued. We also like to assure that all the concerns of the reviewer have been addressed in the best interest of the authors and readers. We have revised the text carefully in the light of the suggestions of the reviewer.

1. My greatest concern is regarding the molar conductivity measurements in DMF. The conductivity values should be dependent on the electrolytic nature of the compounds and independent of the counter anion. The trend in conductivity should be $1:3$ or $3:1 > 1:2$ or $2:1 > 1:1$. See for example, W. J. Geary, *Coord. Chem. Rev.*, 7, 81 (1971). Thus, the trend observed for the complexes is not in line with the principle. Therefore, I suspect that the complexes might contain ionic impurities. The authors have not made any attempt to purify the complexes after synthesis.

Authors' reply: We appreciate the reviewer for this detailed clarification. We have reanalyzed the molar conductivity of complexes using the same solvent (DMF). Here we want to explain that previously we have compared the molar conductivity of complexes

considering per ion in the outer sphere. But now in the revised version, we have discussed the molar conductivity of the whole complexes.

2. What is the likely mechanism of antibacterial and antioxidant activity of the complexes? What is the role of the central metal ion in the overall activity?

Authors' reply: The most of the organic compounds that exhibit only one- or two-dimensional topologies, whereas metal complexes have the potential that they can create three-dimensional structures using coordination chemistry of metal centers. Thus, metal complexes are capable to develop a widespread range of antimicrobial agents. Moreover, metal complexes also have the ability to process antimicrobial actions in a variety of ways such as inhibition of microbial cell wall synthesis, disruption of microbial membrane, inhibition of nucleic acid synthesis, inhibition of protein synthesis, loss of vital substrates, release or exchange of ligands, interruption of membrane functions, denaturation of proteins, redox activation, destruction of DNA, abolishing the enzymatic activities as well as catalytic generation of some toxic species (reactive oxygen species, ROS). Metal ions play significant role in all types of biological activities, as well as their rational design can also be utilized for developing new diagnostic probes or pharmaceutical drugs. The fact that metal atoms can readily give off electrons and produce cations assists their dissolution in biological fluids. Because of their electron deficient nature, they can easily interact with proteins of DNA, which are electron-rich biomolecules. Thus, participating either in the determination/stabilization of their tertiary or quaternary structures or in a catalytic mechanism.

There exist different mechanisms of antioxidant activity depending on the interaction of different ROS with metal complexes as antioxidant agents; interaction either with lipid hydroperoxides or lipid peroxy radicals, thus terminating the basic chain radical process of lipid peroxidation, scavenging of hydroxyl radical as well as the dismutation of superoxide radical anion. The metal ions delocalize the electronic charge distribution of ligands by stabilizing their electronic system. This effect increases the antioxidant profile of ligands.

We have also included this brief explanation about the mechanism of antibacterial and antioxidant activity of the complexes and the role of the central metal ion in the overall activity in the introduction.

3. The antibacterial activity should be quantified by determining the minimum inhibitory concentration (MIC) of the complexes. Besides, studies using cultured bacterial strains are now regarded as preliminary and do not provide a realistic assessment of the antibacterial potency. Bacterial biofilm formation is the main culprit for the development of antibiotic resistance and the severity of most infections that are difficult to treat (Nature Rev. Microbiol., 14, 563 (2016)). So, the compounds should be tested for their ability to inhibit bacterial biofilm formation on model surfaces.

Authors' reply: It would be interesting to explore this aspect. But because of the increasing number of COVID cases and a high mortality rate due to this severe 4th layer, there is a strict lockdown here in Pakistan and here working in our university is almost online and there is no possibility to carry out this test at the moment. Therefore, we are unable to quantify the antibacterial activity by determining the minimum inhibitory concentration (MIC) of the complexes.

4. The antibacterial activity is normally due to the generation of ROS, while the antioxidant activity could be due to the lowering of ROS concentration by the compounds. Thus, a compound with a greater ability to generate ROS would more desirable as an antibacterial agent, while the one with greater ability to lower the concentration of ROS would be more desirable as an antioxidant. Thus, a compound showing both antibacterial and antioxidant activity seems to have a contradictory effect. The authors need to explain this fact.

Authors' reply: Although these mechanisms are considered well-known for both type of activities but as explained in the reply of 2nd comment, compounds exhibit antibacterial and antioxidant activity by different mechanism of actions rather than just generating ROS and lowering concentration of ROS for antibacterial and antioxidant activity, respectively. And there are many natural as well as synthetic compounds reported in the literature that are both antibacterial and antioxidant as well. See the following examples;

- Martelli, G., & Giacomini, D. (2018). Antibacterial and antioxidant activities for natural and synthetic dual-active compounds. *European Journal of Medicinal Chemistry*, 158, 91-105.
- Kalaivanan, C., Sankarganesh, M., Suvaikin, M. Y., Karthi, G. B., Gurusamy, S., Subramanian, R., & Asha, R. N. (2020). Novel Cu (II) and Ni (II) complexes of

nicotinamide based Mannich base: Synthesis, characterization, DFT calculation, DNA binding, molecular docking, antioxidant, antimicrobial activities. *Journal of Molecular Liquids*, 320, 114423.

5. Why did the authors consider Cr(III) as the trivalent metal for their study? In my opinion, Fe(III) and Co(III) would have been more justified considering their bioessential nature. Cr(III) is not a good choice due to the fact that it can generate potentially carcinogenic other Cr species by speciation.

Authors' reply: We would like to clarify that as synthesis was carried out in inert atmosphere using anhydrous chemicals and divalent metal salts of iron and cobalt. As a result, the synthesized complexes are more justified with Fe(II) and Co(II) ions. Furthermore, their colors, magnetic moments and electronic spectra confirmed their divalent nature. We appreciate the reviewer's insightful suggestion regarding the bioessential nature of Fe(III) and Co(III) ions and we will work on it in our next research work.

Cr(III) is a trace element in humans and plays a major role in glucose and fat metabolism. The beneficial effects of Cr(III) in obesity and types 2 diabetes are known. It has been long considered an essential element, but now it has been reclassified as a nutritional supplement. On the other hand, Cr(VI) is a human carcinogen and exposure to it occurs both in occupational and environmental contexts. It induces also epigenetic effects on DNA, histone tails and microRNA; its toxicity seems to be related to its higher mobility in soil and swifter penetration through cell membranes than Cr(III). The dual and opposing roles of this metal make it particularly interesting. A review article with the complete overview of the recent literature intends to underline the important role of Cr(III) for human health and the dangerousness of Cr(VI) as a toxic element.

See the complete review article from following reference;

- Genchi, G., Lauria, G., Catalano, A., Carocci, A., & Sinicropi, M. S. (2021). The double face of metals: The intriguing case of chromium. *Applied Sciences*, 11(2), 638.
6. Establishing the true identity and stability of a metal complex in a solution phase (preferably aqueous buffer such as PBS) is very important for any biological applications. The authors have not made any attempt to study the stability and solution identity of the reported complexes. Elemental analysis data do not provide any indication of the solution

stability of a complex. The stability of a complex (in DMF/PBS) can be qualitatively investigated by UV-visible spectroscopy by monitoring any spectral changes over a reasonable duration (say, up to 48 h).

Authors' reply: The reviewer has raised an important point here. And the authors would like to elaborate this point that in addition to the elemental analysis of complexes, all the complexes have also been analyzed in solution phase. In explanation, we have made the solutions of complexes in DMF solvent and determined their molar conductance values by conductivity meter. Then the next day (after about 20-24 hours), we have qualitatively investigated them by UV-visible spectroscopy. Moreover, two zinc complexes (**7**) and (**14**) have also been analyzed by $^1\text{H-NMR}$ and $^{13}\text{C-NMR}$ spectra using their solutions in DMSO.

7. The authors have presented the MS data only for the ligands. Why not for the complexes? The measurement of ESI-MS for the complexes and the observation of a prominent peak corresponding to the complex would also tell whether the complex is stable in a solution phase or not.

Authors' reply: Thank you for this suggestion. But since educational institutions are closed due to the current situation of COVID-19, we have not yet been able to incorporate the ESI-MS for the complexes. But to study the stability of complexes in solution phase, we have included the $^1\text{H-NMR}$ and $^{13}\text{C-NMR}$ spectra of two zinc complexes (**7**) and (**14**). And conductance and UV-Vis analysis of all the complexes are already included in the manuscript.

8. I have concerns regarding the stability of the VO(IV) complexes 1, 8, and 15. The two ligands are shown to bind the oxovanadium moiety as bidentate ligands. But, the ligand is forming a 3-membered ring. In coordination chemistry, 3-membered rings are generally strained and lead to the formation of unstable chelates unless stabilized by strong and special electronic factors.

Authors' reply: We agree with the reviewer that 3-membered ring systems are generally more strained and least stable. But herein this manuscript, our VO(IV) complexes (**1**), (**8**), and (**15**) are 4-membered ring systems, which are quite stable and less strained than that of 3-membered ring systems.

9. Did the authors try to obtain the single-crystal X-ray diffraction structures of the complexes? It is a powerful and valuable tool in coordination chemistry. I strongly suggest that the authors should always try to report crystal structures (at least one or two if not all) in their future manuscripts.

Authors' reply: We have tried our best to grow the single crystals of the respective metal complexes by changing different solvent ratios and other conditions. Unfortunately, we couldn't succeed to get fine single crystal from the powder metal complexes. Then, all the ligands (Z₁)-(Z₃) and their selected metal complexes have been optimized by DFT to insight their geometries in absence of their well resolved SC-XRD data.

10. NMR data of the ligands: please provide the coupling constant (J) values in the data section.

Authors' reply: We have made the suggested change and now coupling constant (J) values have been added in the ¹H-NMR data section.

11. Please provide the DFT coordinates of the optimized structures in the supporting information. Besides, the HOMO/LUMOs of the remaining compounds should be provided as supporting information.

Authors' reply: We agree with the reviewer and in response of this comment, we have included DFT coordinates of the optimized structures in the supporting information as Tables S7-S12. We have also summarized the optimized bond lengths and bond angles of studied compounds in Table S13. Moreover, the HOMO/LUMOs of all the other compounds have also been provided in the supporting information as Figures S23-S25.

12. Antibacterial study: The zone-inhibition images for all the compounds should be provided as supporting information.

Authors' reply: We have added zone-inhibition images of representative compounds against each bacterial strain in the supporting information as Figure S28.

13. Figures 6 & 7: The error bars are really missing and should be added.

Authors' reply: Thank you for pointing out this. We have revised both the suggested figures (6 & 7) by adding error bars and presented figures are now more accurate.

14. The graphical abstract looks are clumsy to me. The authors could present a simplified one by considering only three aspects, namely, DFT, antibacterial, and antioxidant studies. Similarly, the title of the manuscript could be improved.

Authors' reply: We agree and made the suggested change by revising the graphical abstract and title of the manuscript considering only three aspects; DFT, antibacterial, and antioxidant studies.

15. I suggest that the authors go for one round of improvement of the English used in the manuscript in order to improve the readability.

Authors' reply: The overall quality of the manuscript and the English language has been carefully improved with the help of English expert.

Reviewer: 2

Comments to the Author(s)

The synthesis of Schiff bases from the condensation of 2-amino-6-ethoxybenzothiazole or 2-amino-6-nitrobenzothiazole with 5-chloroisatin or isatin and their 3d-transition metals complexes is reported in this ms. The compounds were characterized by spectroscopic methods (UV-Vis, FT-IR, ¹H-NMR, ¹³CNMR), mass spectrometry, elemental analysis, and by computational calculations employing the B3LYP/6-31+G(d,p) functional of DFT. The in vitro antibacterial potential of the compounds is also reported. The manuscript can be published in Royal Society Open Science only after major revision and reconsideration.

Authors' reply: We are grateful to the reviewer for the constructive comments about the paper. A major revision of the paper has been carried out to take all of them into account. And in the process, we believe the paper has been significantly improved.

1. The crystal structures would be supportive for the characterization of the 24 compounds. Was any crystallization be attempted?

Authors' reply: Initially, the precipitates were recrystallized using equimolar ratio of ethanol and ether to get pure product. After that to get the crystals, we have tried some more solvent mixtures varying their ratios, but we couldn't succeeded to get fine single crystal. Then, all the ligands (**Z₁**)-(**Z₃**) and their selected metal complexes have been optimized by DFT to insight their geometries in absence of their well resolved SC-XRD data.

2. The structural and biological study of the Schiff base derived from 2-Amino-6-methoxybenzothiazole with Isatin and its metal complexes has been already published by Hassan et.al. (2020 International Journal of Pharmaceutical Research 12(1), pp. 323-336). This work should be taken into consideration during the preparations of this manuscript. The Schiff base of 5-chloro isatin with 2-amino 5-nitrobenzothiazole and its complexes with ZnCl₂ and CuCl₂ are already reported by Shakir M et.al in Journal of Photochemistry and Photobiology B: Biology Volume 157, Pages 39 – 56, 2016. Please rationalize the repetition of their quoting here. Similarly, the spectroscopic data have already reported.

Authors' reply: We would like to clarify here that we have synthesized three aminothiazole Schiff base; (**Z₁**) from 2-amino-6-ethoxybenzothiazole and isatin, (**Z₂**) from 2-amino-6-ethoxybenzothiazole and 5-chloroisatin, and (**Z₃**) from 2-amino-6-nitrobenzothiazole and 5-chloroisatin. Our aminothiazole moieties are different from that used in the recommended papers. We have not used both of the above mentioned aminothiazole (2-amino-6-methoxybenzothiazole and 2-amino-5-nitrobenzothiazole) in the synthesis of our Schiff bases. Then how we can repeat as well as report their spectroscopic data.

3. Please check the electric charges balance of the formulae shown in Scheme 1. E.g Cr(III) needs 3 CH₃COO⁻ anions and not one as it is shown. There are two Cl⁻, counter anions in the case of Co(II). Similarly Co(II), Cu(II) and Ni(II) dichloride.

Authors' reply: Thank you for this detailed clarification. Revised scheme specifying the number of counter anions [three acetate ions for Cr(III), two chloride ions for Co(II), Ni(II), Cu(II) and one sulphate ion for VO(IV), Fe(II), Zn(II)] have been included in the manuscript.

4. Please keep the chemical nomenclature. E.g In IR there are “vibrational” bands. NMR resonance signals not “peaks”, “peaks for the methylene”, “carbon C4 peak” etc. There are molecular fragments or ions in Mass spectrometry and not “peaks” etc

Authors' reply: We agree with the reviewer's assessment. Accordingly, throughout the discussion of IR, NMR and mass spectrometry, we have corrected ‘peaks’ as ‘vibrational bands’, ‘signals’ and ‘molecular fragments’, respectively.

5. Infra-red and NMR Spectra

A table with the characteristic vibrational bands and NMR signals of the compounds should be added in the text.

Authors' reply: A table listing the characteristic vibrational bands of all compounds is already given in the Supplementary Information as Table S2. But as suggested by the reviewer, the proton and carbon NMR signals of ligands (**Z₁**)-(**Z₃**) as well as their diamagnetic Zn(II) complexes are now given as Table S3-S4 and S5-S6, respectively.

6. Molar conductance

Please check the consistence of the Molar conductance values (in table S2) and the formula of the compounds given in table S1. E.g. DMF solutions of electrolyte with 2 ions exhibit molar conductance value 65-90, while the corresponding value for the solution of electrolyte with 3 ions is 130-170 $\Omega^{-1}\text{cm}^2\text{mol}^{-1}$. The values measured correspond to 2 ions electrolytes which is not the case of proposed formulae in the case of Cr(III) acetate, copper dichloride, nickel dichloride cobalt dichloride etc.

Authors' reply: Thank you for this detailed explanation. We have again performed the molar conductance measurement for complexes using the same solvent (DMF). And now we have compared the molar conductance values of complexes considering all the electrolytes. But in the previous version, we have discussed the molar conductivity of complexes considering per ion in the outer sphere.

7. Antibacterial activity

Please give the diameter of the paper used in the inhibition zones (IZ). How the IZ diameter of e.g 4 mm in case of Z1 is measured?

Authors' reply: The reviewer has raised an important point here. The diameter of the paper used in the inhibition zones (IZ) is 6 mm. As we have mentioned in the experimental detail of antibacterial activity that “At the end, clear or inhibition zones were noted (in millimeter) for all the tested compounds and standard drug against each bacterial strain.” We have not compared the overall diameter of the inhibition zones. We have only discussed the clear zone which is definitely the actual area representing the inhibitory activity of compounds.

In the case of (Z1), the diameter of inhibition zone was 10 mm, but the clear zone area was 4 mm (after subtracting the diameter of paper as shown in Figure S28).

8. Figure 6. The negative values in the histograms of the antibacterial activity of aminothiazole ligands (Z1)-(Z3) vs their transition metal complexes (1)-(21) is not clear. Please explain.

Authors' reply: The negative values for diameter of inhibition zone in Figure 6 represent the antibacterial activity of all the compounds against Gram-negative bacterial strains

(*Escherichia coli* and *Salmonella typhimurium*). Actually, the inhibition values were not negative, but these are graphically represented as negative for Gram-negative bacteria in Figure 6.

At the end, we would like to thank both the reviewers again for taking the time to review our manuscript and helping us to improve the consistency of our manuscript. We have been able to incorporate changes to reflect most of the suggestions provided by the reviewers. In addition to the above comments, all spelling and grammatical errors have been corrected throughout the manuscript. And overall, the quality of the manuscript has been enhanced. We hope that the manuscript would now be suitable for acceptance and publication in *Royal Society Open Science*. I look forward hearing your positive response.

Profound regards,

Dr. Sajjad Hussain Sumrra
Tenured Associate Professor
(Corresponding author)